# Heparan sulfate regulates amphiregulin programming of tissue reparative lung mesenchymal cells during influenza A virus infection in mice

Lucas F. Loffredo[1], Anmol Kustagi[1], Olivia R. Ringham[1], Fangda Li[1], Kenia de los Santos-Alexis[1], Anjali Saqi [2] & Nicholas Arpaia [1,3] ✉

Amphiregulin (Areg), a growth factor produced by regulatory T (Treg) cells to facilitate tissue repair, contains a heparan sulfate (HS) binding domain. How HS, a highly sulfated glycan subtype that alters growth factor signaling, influences Areg repair functions is unclear. Here we report that inhibition of HS in various cell lines and primary lung mesenchymal cells (LMC) qualitatively alters Areg downstream signaling. Utilization of a panel of cell lines with targeted deletions in HS synthesis–related genes identifies the glypican family of HS proteoglycans as critical for Areg signaling. In the context of influenza A virus (IAV) infection in vivo, an Areg-responsive subset of reparative LMC upregulate glypican-4 and HS; conditional deletion of HS primarily within this LMC subset results in reduced repair characteristics following IAV infection. This study demonstrates that HS on a specific lung mesenchymal population is a mediator of Treg cell–derived Areg reparative signaling.

In order to engage in their myriad functions, cells of the immune system must migrate through, work within, and communicate with dense tissue networks of non-immune cells. While the implications of these interactions are well-studied in the context of anti-pathogen responses, our knowledge of the role these interactions play in regulating tissue homeostasis and repair remains far more limited.

One mechanism by which immune cells mediate tissue regulatory changes is via the production of amphiregulin (Areg). Areg is a growth factor of the epidermal growth factor receptor (EGFR) ligand family that has been identified as an immune-derived mediator of repair in multiple tissue damage and disease contexts[1]. Previous reports have demonstrated that lung regulatory T (Treg) cells produce Areg in the context of influenza A virus (IAV) infection in mice and that genetic deletion of *Areg* specifically in Treg cells impairs proper recovery of blood oxygen saturation, a feature found to be independent of the canonical role Treg cells play in immunosuppression[2]. While these

tissue effects were initially thought to be the result of Areg signaling directly to epithelial cells, further work from our group demonstrated that a distinct subset of lung mesenchymal cells (LMC) are in fact the critical Areg-sensing intermediate in this context[3]. This LMC population showed high EGFR expression, unique responsiveness to Areg, and spatial localization near areas of alveolar damage, characteristics that endow this subset with the ability to send reparative signals to epithelial cells in response to Treg cell-derived Areg. We and others have referred to this cell type as "*Col14a1*+ lung mesenchymal cells" (abbreviated here as "Col14-LMC")[4], based on high expression of *Col14a1* (encoding Type XIV collagen); transcriptionally similar LMC populations have been referred to in other publications as mesenchymal alveolar niche cells or adventitial fibroblasts[5–8]. Our previous work also included comparisons to the other two most sizable subsets of mouse LMC, marked by expression of *Col13a1* ("Col13-LMC") or *Hhip* ("Hhip-LMC"), alternatively referred to in the literature as alveolar

[1]Department of Microbiology & Immunology, Columbia University, New York, NY, USA. [2]Department of Pathology and Cell Biology, Columbia University Irving Medical Center, New York, NY, USA. [3]Herbert Irving Comprehensive Cancer Center, Columbia University, New York, NY, USA. ✉e-mail: na2697@cumc.columbia.edu

fibroblasts and peribronchial fibroblasts, respectively[7]; these LMC populations expressed lower amounts of EGFR and were less responsive to Areg stimulation.

Heparan sulfate (HS) is a type of glycosaminoglycan (GAG) that is produced and presented by tissue cells (see Fig. 1A for a visual summary of HS biology to accompany this description). Among the diverse types of GAGs, HS has received increased attention due to its ability to

harbor the greatest amount of sulfate groups, giving it a highly negative charge[9] and allowing it to affect the signaling of a diverse array of positively-charged HS-binding proteins (HSBPs)[10]. HS is produced and presented in a protein-bound form, either on cell surface proteins (syndecans, which have a transmembrane domain, or glypicans, which are glycosylphosphatidylinositol [GPI]-anchored) or on secreted extracellular matrix (ECM) proteins[9]. HS is initially synthesized in the

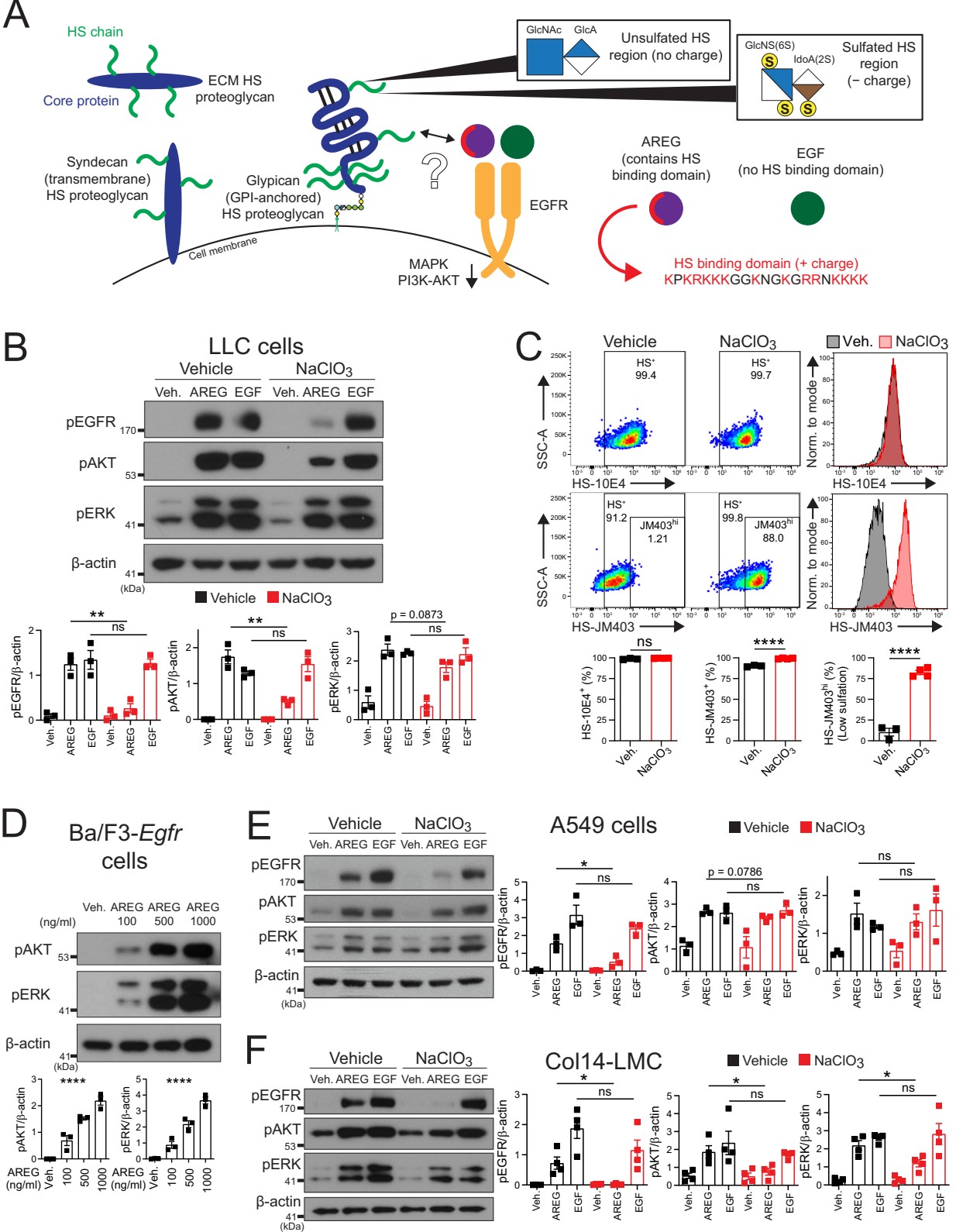

**Fig. 1 | Areg signaling is altered but not abrogated in the context of HS inhibition. A** Diagram of different types of heparan sulfate (HS)-presenting proteoglycans, heterogeneity in sulfation along a single HS chain, and epidermal growth factor receptor (EGFR) ligand interactions with HS. ECM extracellular matrix, GlcNAc N-acetylglucosamine, GlcA glucuronic acid, GlcNS(6S) N-sulfoglucosamine (6-O-sulfated), IdoA(2S) iduronic acid (2-O-sulfated), AREG amphiregulin, EGF epidermal growth factor. **B** Western blotting for phospho-EGFR (Y1068), phospho-AKT (S473), phospho-ERK (T202/Y204), and β-actin of vehicle or sodium chlorate (NaClO$_3$)-treated (16–18 h) LLC cells, stimulated (15 min.) with vehicle, mouse AREG (500 ng/ml), or mouse EGF (100 ng/ml). Representative western blots shown. $n = 3$ per condition, graphs contain all values from three experiments. **C** Flow cytometry using HS-directed antibodies 10E4 or JM403 on LLC cells treated with vehicle ($n = 3$) or NaClO$_3$ ($n = 4$) (16–18 h). Representative flow cytometry plots shown. Gating based on fluorescence-minus-one (FMO) controls. Percent staining positive displayed in plots. Graphs contain all values from two experiments. **D** Western blotting for phospho-AKT (S473), phospho-ERK (T202/Y204), and β-actin of vehicle or mouse AREG-stimulated (at various concentrations) (15 min.) Ba/F3-*Egfr* cells. Representative western blots shown. $n = 3$ per condition,

graphs contain all values from three experiments. **E** Western blotting for phospho-EGFR (Y1068), phospho-AKT (S473), phospho-ERK (T202/Y204), and β-actin of vehicle or NaClO$_3$-treated (16–18 h) A549 cells, stimulated (15 min.) with vehicle, human AREG (500 ng/ml), or human EGF (100 ng/ml). Representative western blots shown. $n = 3$ per condition, graphs contain all values from three experiments. **F** Western blotting for phospho-EGFR (Y1068), phospho-AKT (S473) phospho-ERK (T202/Y204), and β-actin of vehicle or NaClO$_3$-treated (16–18 h) Col14-LMC, stimulated (15 min.) with vehicle, mouse AREG (500 ng/ml), or EGF (100 ng/ml). Col14-LMC were gated/sorted as shown in Supplementary Fig. 2 (negative bead enriched), then at ~24 h post-plating non-adherent cells were washed away/media changed, with subsequent vehicle/NaClO$_3$ treatment. Representative western blots shown. $n = 4$ per condition, graphs contain all values from four experiments. Statistical analysis done for western blot/flow cytometry data where two groups were compared using two-tailed unpaired *t*-tests, and for western blot data where four groups were compared using one-way ANOVA. Mean and standard error displayed on graphs; ns not significant, *$0.01 < p < 0.05$, **$0.001 < p < 0.01$, ****$p < 0.0001$. Source data are provided as a Source Data file.

Golgi apparatus as an unsulfated polysaccharide chain consisting largely of repeating glucuronic acid and N-acetylglucosamine disaccharide subunits; this chain is then altered by sulfate group addition during Golgi apparatus trafficking by various sulfotransferase enzymes, which impart its negative charge[9]. Significant heterogeneity exists along each individual HS chain, with different areas containing low or high levels of sulfation[9]. From past work on HS, the paradigm has emerged that HS is ubiquitously expressed in all tissue cells[11]; however, most reports studying HS in tissue environments do not explicitly evaluate levels of HS on various tissue cell types.

Unlike the canonical EGFR ligand family member epidermal growth factor (EGF), Areg contains an HS-binding domain. While early reports found that Areg is dependent on HS for signaling in vitro[12,13], few studies have investigated interactions between Areg and HS since this initial work. Interestingly, while there are numerous cellular sources for Areg (e.g., epithelial cells[14]), only Areg from certain immune cells in specific disease contexts effectively supports tissue repair, implying a highly spatially localized component for its tissue regulatory roles. We reasoned that due to this apparent spatial specificity of action for Areg, and since HS expression varies greatly among tissue cell types in settings of inflammation, there may be unappreciated aspects of Areg–HS interactions in these processes that merit further investigation.

In this report, we sought to broaden the investigation of Areg–HS interactions by using chemical and genetic inhibition of HS in vitro, in order to analyze downstream signaling, transcriptional effects, and molecular components of this relationship. We further these inquiries to an in vivo tissue context using models of lung damage and conditional HS deletion mice targeting a specific subpopulation of LMC. Furthermore, we find that both a specific core protein for HS presentation (glypican-4) and HS itself are uniquely upregulated on Areg target cells in the lung (Col14-LMC) during IAV infection, endowing them with a heightened ability to respond to reparative Treg cells. Our findings demonstrate a role for HS on a distinct subset of LMC, which is upregulated and utilized in order to respond to immune cell-derived Areg and enact proper tissue repair modalities following IAV-induced damage.

## Results

### Areg signaling is altered but not abrogated in the context of HS inhibition

To confirm and extend earlier observations regarding the dependency of the Areg HS-binding domain for signaling (Fig. 1A)[12,13], we performed experiments using the Lewis lung carcinoma (LLC) mouse epitheloid cancer cell line, which has high expression of HS and EGFR[15,16], using known HS signaling inhibitors. Notably, while Areg signaling dependency on HS was previously only shown using an

indirect readout of DNA synthesis, we chose to test this molecularly by assessing the phosphorylation of EGFR via western blotting in response to ligand stimulation. EGF, which lacks an HS-binding domain, was used as a control. The dependency on HS for Areg-, but not EGF-signaling, was shown using three HS-altering treatment methods: (1) sodium chlorate (NaClO$_3$), which prevents formation of sulfate donors in the cytoplasm, thus reducing sulfation in displayed HS[12] (Fig. 1B), (2) heparinase I/III, which enzymatically cleaves HS from the surface of cells[12] (Supplementary Fig. 1A), and (3) surfen, a positively-charged small molecule antagonist of HS which functions by binding up negatively charged sulfated regions of HS to make them refractory to HSBP engagement[17] (Supplementary Fig. 1B).

To assess the effects of these HS-altering treatments on LLC cell surface-bound HS, we used two established antibodies for HS assessment: 10E4, which targets highly sulfated regions on HS[18], and JM403, which targets regions of low sulfation on HS[19]. As visualized by 10E4 staining, NaClO$_3$ treatment did not prevent the production and presentation of HS by LLC cells; in fact, enough sulfated HS regions are maintained on cells in this context to prevent any reduction in 10E4 staining (Fig. 1C). However, staining with JM403 shows a full shift in the profile of LLC cells from a JM403$^{lo}$ to JM403$^{hi}$ phenotype upon NaClO$_3$ treatment (Fig. 1C). This is an indication that low-sulfation regions of HS predominate upon treatment with NaClO$_3$. Heparinase I/III treatment of LLC cells resulted in reduced staining for 10E4 and JM403 (Supplementary Fig. 1C). Surfen treatment did not alter staining for 10E4 or JM403 on LLC cells (Supplementary Fig. 1D). To summarize, here we demonstrate that HS inhibition with NaClO$_3$ leaves HS moieties intact but induces a dramatic shift in their sulfation profile, while heparinase I/III appears to mediate cleavage of HS from the cell surface, and surfen seems to not affect the overall profile of HS but instead alters the availability of positively-charged HSBP binding sites. Since heparinase I/III cleavage of HS leaves residual fragments in the surrounding media that can bind up HSBPs upon subsequent treatment (thus potentially confounding experimental results), and since surfen treatment has been shown to induce cell death[20], we focused on NaClO$_3$ treatment as the cleanest experimental method for chemical HS inhibition in subsequent experiments.

Downstream signaling via EGFR occurs through the MAPK and the AKT-mTOR pathways[21]. We evaluated signaling via these pathways from Areg and EGF treatment with or without HS sulfation inhibition (NaClO$_3$ treatment) using western blotting, expecting this to mirror the full inhibition seen for EGFR phosphorylation. Instead, we found that in the context of HS sulfation inhibition (NaClO$_3$) in Areg-stimulated cells, ERK phosphorylation (MAPK pathway) is maintained at similar levels, while AKT phosphorylation is significantly reduced but still present, compared to vehicle controls (Fig. 1B). Prior research investigating Areg signaling dependence on HS has implied that

without HS, Areg is fully deficient in signaling; our findings run counter to this, indicating that there are HS-dependent and HS-independent modalities of Areg signaling.

To eliminate the possibility that residual MAPK and AKT signaling in the setting of NaClO$_3$ treatment is due to impartial HS inhibition, we tested the ability of Areg to signal in Ba/F3 cells, a mouse pro-B cell line reported to completely lack HS[22]—which we confirmed using HS-targeting antibodies (Supplementary Fig. 1E). This cell line also lacks expression of EGFR[23]; thus, we transiently transfected a construct encoding mouse *Egfr* into these cells and confirmed its signaling capability via treatment with EGF (Supplementary Fig. 1F). Upon treatment with Areg, we saw a dose-dependent response in downstream ERK and AKT phosphorylation in these cells (Fig. 1D) (EGFR phosphorylation was not able to be assessed in this cell type). The dose dependency of this response indicates that Areg is binding and inducing bona fide activation of its receptor, in a context completely devoid of HS.

An additional method we utilized to test the effects of HS on Areg signaling was to use a pre-binding strategy. Recombinant Areg was first incubated with free HS to block available HS-binding sites prior to applying this mixture to LLC cells so that any residual signaling toward EGFR must occur independently of this binding domain. Using this strategy, we found that EGFR phosphorylation and AKT phosphorylation were significantly reduced, but ERK phosphorylation remained fully intact (Supplementary Fig. 1G). These results further confirmed that the HS-binding domain alters Areg signaling, primarily affecting the AKT-mTOR pathway rather than the MAPK pathway.

To test that this is also relevant for human Areg, we performed select experiments on A549 cells, a human adenocarcinoma alveolar epithelial cell line with ubiquitous expression of HS and high expression of EGFR[24,25]. NaClO$_3$ treatment largely recapitulates our previous results seen in LLC cells upon Areg and EGF stimulation—EGFR phosphorylation is significantly reduced, AKT phosphorylation shows a trend towards reduction but is still present, and ERK phosphorylation maintains full signaling potential (Fig. 1E). Similarly, NaClO$_3$ treatment of A549 cells promotes a wholesale shift of JM403 staining from a low- to high-staining profile (Supplementary Fig. 1H). Thus, the HS-dependent vs. -independent signaling modalities of Areg are also present in the human context.

Lastly, we investigated whether a similar signaling pattern is present in primary cells relevant for Areg signaling; we chose to use Col14-LMC for this cell type, given their EGFR^hi phenotype and reparative interaction with Treg cell-derived Areg described in our previous publication[3]. In our prior study, we demonstrated the ability to isolate and culture the major subsets of lung mesenchymal cells described above (Col14-LMC, Col13-LMC, and Hhip-LMC), via flow cytometric sorting from enriched lung mesenchyme (Supplementary Fig. 2)[3]. Upon sorting and culturing of Col14-LMC, we found that stimulation with Areg in the presence of NaClO$_3$ treatment resulted in significantly decreased EGFR and AKT phosphorylation, similar to that seen in LLC cells (Fig. 1F). For MAPK pathway signaling, a significant decrease in ERK phosphorylation was observed for Areg stimulation in the presence of NaClO$_3$; however, an increase in ERK phosphorylation was apparent when compared to NaClO$_3$ treatment alone (Fig. 1F), confirming that Col14-LMC are able to undergo HS-independent signaling as described above for LLC. We also observed a similar surface HS pattern as was observed in LLC when Col14-LMC were stained with HS-targeted antibody JM403 in the presence of NaClO$_3$ treatment (shift from JM403^lo to JM403^hi) (Supplementary Fig. 1I). Based on these observations, we concluded that the HS-dependent and -independent signaling modalities are preserved in this Areg signaling-relevant primary cell type.

## Transcriptional profile of HS-dependent vs. -independent Areg signaling in Col14-LMC

We reasoned that HS-dependent vs. -independent signaling may impact the transcriptional signature of target cells. To this end, we performed bulk RNA-seq on vehicle-treated/Areg-stimulated vs. NaClO$_3$-treated/Areg-stimulated Col14-LMC. Areg treatment was administered at three different concentrations; this was done to allow for comparison between similarly sized gene signatures for HS-unaltered vs. HS sulfation-inhibited conditions, in the case that the latter resulted in overall lessened gene expression changes in response to Areg stimulation. Observing total numbers of up/downregulated differentially expressed genes (DEGs) in these conditions (Fig. 2A), this was indeed the case, with NaClO$_3$-treated Col14-LMC showing lower DEGs compared to vehicle-treated at each concentration of Areg tested. However, HS sulfation-inhibited Col14-LMC still exhibits a dose-responsive increase in DEGs in response to increasing Areg concentrations, with an appreciable gene signature of 2504 total DEGs at the highest concentration, confirming our previous observation that Areg is able to signal in the absence of proper HS engagement. Principle component analysis of full gene expression profiles identified Areg responsiveness as the primary driver of separation between samples, with little separation seen on the secondary axis between vehicle and NaClO$_3$-exposed samples, consistent with an evident but smaller (in comparison to Areg treatment) effect of NaClO$_3$ exposure at baseline (1059 DEGs) (Fig. 2B).

In order to compare HS-intact and HS sulfation-inhibited cells at similar overall numbers of identified DEGs, we performed pathway analysis (KEGG pathways) on upregulated genes from the lowest Areg concentration for vehicle-treated Col14-LMC (200 ng/ml; 1459 upreg. DEGs) and the highest Areg concentration for NaClO$_3$-treated Col14-LMC (1000 ng/ml; 1285 upreg. DEGs). Significantly upregulated pathways for each are summarized in Fig. 2C; pathways upregulated in both contexts (middle of Venn diagram) are HS-independent (i.e., they are upregulated regardless of HS sulfation inhibition), while pathways upregulated only in the HS-intact context (left of Venn diagram) are HS-dependent. There were no significantly upregulated pathways represented in the HS sulfation inhibition context that were not represented in the HS-intact context. Strikingly, mirroring our previous results at the signaling level, the MAPK signaling pathway is significantly upregulated in both contexts (HS-independent), while the ERBB signaling pathway and the PI3K-AKT signaling pathway are only upregulated in the HS-intact context (HS-dependent).

Genes involved in tissue/growth factor signaling and angiogenesis (e.g., *Il11*, *Lif*, *Vegfa*, *Angtl4*, *Tgfb1*, *Bmp2*) show significant upregulation across increasing Areg concentrations in both HS-intact and sulfation-inhibited conditions, with similar or only slightly reduced levels of induction in the latter (Fig. 2D), demonstrating that these tissue modalities appear to be largely independent of HS involvement. Other pathways were only substantially upregulated across groups in the HS-intact scenario (HS-dependent), including genes related to the Hippo pathway (*Ccnd1*, *Ccn2*), actin mobilization (*Actg1*), and intermediate filament induction (*Des*, *Vim*) (Fig. 2E). These modalities have the potential to regulate cell fate, cell polarity, and movement in response to an activating stimulus[26]. Interestingly, *Gjb2*, the gene encoding connexin-26, shows this pattern as well; this gene is critical for the proper formation of gap junctions[27], which may point to a role in fostering intercellular communication as an additional HS-dependent function.

Fibroblast growth factor (FGF) family members and Wnt ligands have been previously described as mediators of mesenchymal cell-mediated repair from lung damage[28–30]. Thus, we assessed the patterns of expression across Areg concentrations in HS-intact and HS-inhibition conditions for FGF and Wnt genes, showing here all such genes expressed on average >2 TPM (Supplementary Fig. 3). Notably, *Fgf1*, *Fgf18*, *Wnt2*, and *Wnt2b* show similar patterns to the HS-dependent genes from Fig. 2—substantial upregulation by Areg treatment in the HS-intact scenario and lessened or absent upregulation upon HS inhibition. While this same pattern is not apparent for all FGF and Wnt genes, the HS-dependent nature of certain of these genes may

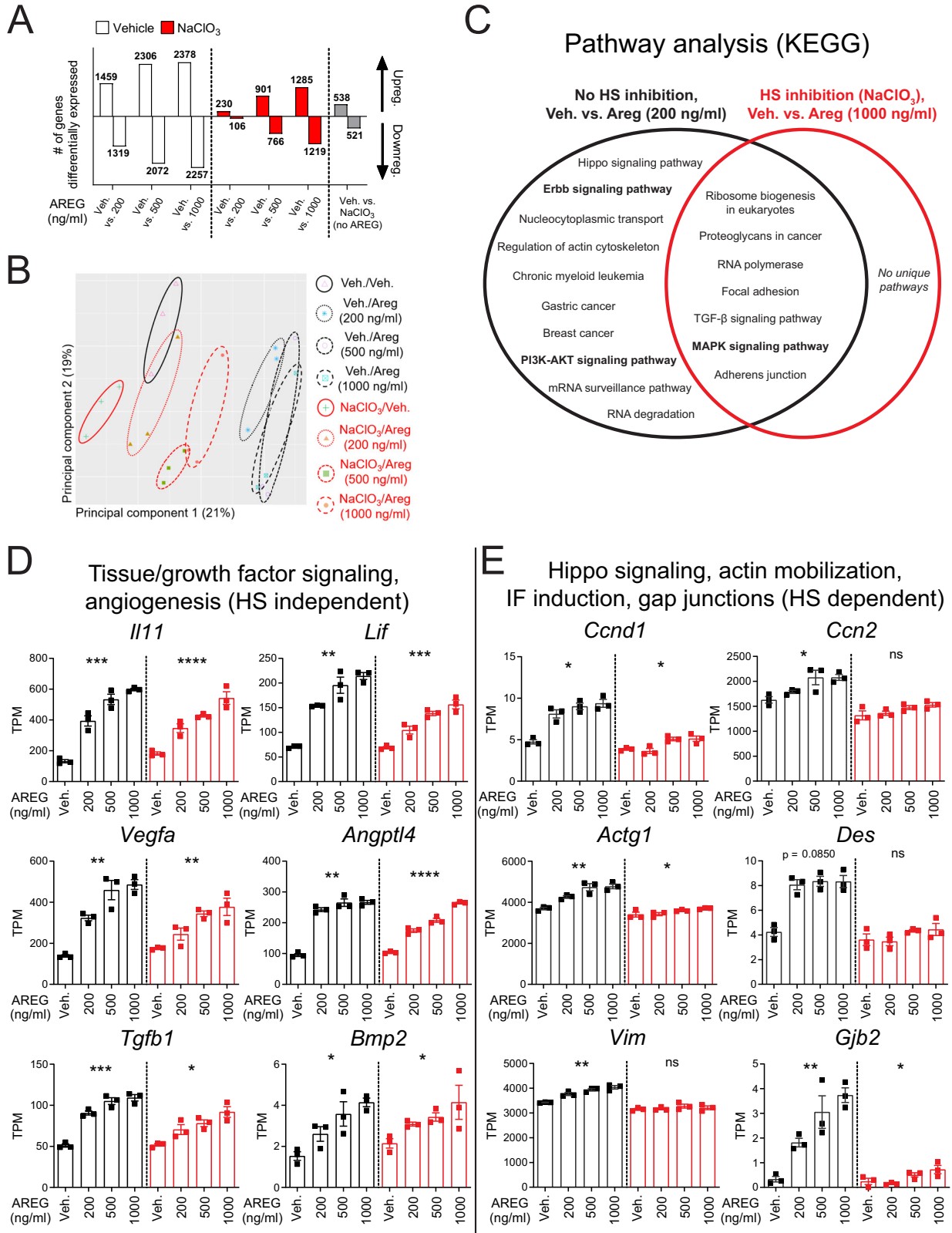

point to a specific set of reparative mediators induced by Areg that require HS on target cells for proper induction.

### HS-related gene knockout cell lines identify glypican-1 as a critical HS core protein for Areg signaling

We next sought to understand the specific characteristics of HS that confer its ability to interact with Areg. The genes encoding mediators

of HS synthesis and modification are summarized in Supplementary Fig. 4[9]. To ascertain which of these HS-related genes are critical for Areg signaling, we created a panel of CRISPR-Cas9-mediated knockout (KO) LLC cell lines. We began by querying a publicly available RNA-seq dataset of WT LLC cells to get a baseline layout of HS-related gene expression in this cell line (Supplementary Fig. 5A)[31]. We chose to include enzymes involved in HS extension and sulfation, as well as

**Fig. 2 | Transcriptional profile of HS-dependent vs. -independent Areg signaling in Col14-LMC.** Bulk RNA-seq of Col14-LMC treated with vehicle or NaClO$_3$ (16–18 h), then stimulated with varying concentrations of mouse AREG for 4 h. Col14-LMC were gated/sorted as shown in Supplementary Fig. 2 (negative bead enriched), then at 24 h post-plating non-adherent cells were washed away/media changed, with subsequent vehicle or NaClO$_3$ treatment (16–18 h) prior to AREG stimulation. **A** Significant differentially expressed genes determined by DESeq2 (FDR-adjusted *p* value < 0.05, no FC cutoff), for comparisons indicated on *x*-axis. **B** Principal components analysis of all samples from bulk RNA-seq of Col14-LMC treated with vehicle or NaClO$_3$ (16–18 h) and stimulated with varying concentrations of mouse AREG. **C** Pathway analysis (gProfiler, KEGG pathways) on upregulated genes from RNA-seq, from the "No HS inhibition, vehicle vs. AREG (200 ng/ml)" comparison (1459 DEGs) (black circle), and the "HS inhibition, vehicle vs. AREG (1000 ng/ml)" comparison (1285 DEGs) (red circle). Certain pathways highlighted in the "Results" section boldened. **D, E** Transcript per million (TPM) values from RNA-seq for representative genes from select functional pathways (*n* = 3 per group). IF intermediate filament. Statistical analysis done for post hoc RNA-seq data of individual genes across four groups using Kruskal–Wallis tests. Mean and standard error displayed on graphs; ns not significant, *0.01 < *p* < 0.05, **0.001 < *p* < 0.01, ***0.0001 < *p* < 0.001, ****p* < 0.0001. Source data are provided as a Source Data file.

certain core proteins that harbor HS (Fig. 3A), while avoiding genes involved in HS chain initiation due to their myriad roles in forming other types of glycans and additional cellular processes[9]. Two separate single-cell-derived clonal lines were generated for each targeted gene, with the exception of *Sdc1*, where we were only able to generate one subline. For all KO sublines, we confirmed a reduction in expression at the mRNA level (Supplementary Fig. 5B), and modification of HS (for enzymes) or reduction in core protein levels (as discussed below).

*Ext1* and *Ext2* are non-redundantly responsible for HS chain extension (and are not involved in the formation of other glycans); thus, we anticipated that the KO of these genes would result in full HS ablation. Accordingly, upon knockout of *Ext1* or *Ext2* in LLC cells, we saw an elimination of staining for HS-directed antibodies 10E4 and JM403 (Supplementary Fig. 5C). As expected, *Ext1*- and *Ext2*-KO sublines show a profound reduction in Areg signaling induction compared to WT controls, as quantified by EGFR phosphorylation relative to EGF treatment (to account for differences in EGFR levels between lines) (Fig. 3B).

To alter the sulfation of HS, we chose to delete *Ndst1*, *Hs2st1*, and *Hs6st1*—highly expressed representative genes of the N-sulfotransferase, 2-O-sulfotransferase, and 6-O-sulfotransferase families responsible for sulfation of different sites on HS. While western blots attempting to demonstrate protein-level depletion in sulfation enzyme KO sublines were unclear, significant alterations were viewed in the HS profile of these sublines using 10E4 and JM403 antibodies: *Ndst1*-KO sublines show a reduction in 10E4 staining and an increase in JM403 staining; *Hs2st1*-KO sublines show a reduction in JM403 staining; and *Hs6st1*-KO sublines show an increase in 10E4 and JM403 staining (Supplementary Fig. 5D). Based on prior paradigms in the field indicating that sulfation state of HS is the determinative factor in its ability to bind HSBPs[10], we expected that KO of the sulfotransferase enzymes would be effective in altering Areg signaling; however, we found that none of the sulfotransferase enzyme KO sublines exhibited reduced Areg signaling compared to WT controls (Fig. 3C).

To determine why deletions in sulfotransferase genes did not affect Areg signaling, we engaged in another line of experimentation to test the ability of Areg to bind to HS with altered sulfation patterns. Heparin is a highly sulfated subtype of HS that is widely used as a clinical anticoagulant. Using a similar Areg pre-binding strategy as in Supplementary Fig. 1H, we tested the ability of different heparin variants, chemically modified to lack sulfation at certain sites, to bind Areg and inhibit EGFR signaling. As expected, we found that Areg pre-bound to heparin was unable to induce signaling (Fig. 3D). Of the differentially desulfated heparin variants, only N-desulfated heparin showed reduced binding to Areg, as demonstrated by the similar ability of Areg to induce signaling when pre-incubated with this variant compared to vehicle-exposed Areg (Fig. 3D). This finding could be taken as evidence that sulfation at the N-site is important for Areg binding to heparin and/or HS; however, in light of the fact that Areg signaling is inhibited by pre-incubation with a reacetylated form of N-desulfated heparin, we believe it more likely that a residual positive charge on N-desulfated heparin (prior to re-acetylation) is leading to repulsive ionic forces that are weakening Areg binding to this variant. Thus, when taken together

with the results from our panel of sulfotransferase KO clones (Fig. 3C), these data indicate that Areg does not show a strong preference for specific sulfation modifications in its binding to HS, rather the net negative charge of HS mediates its interaction with Areg.

With regards to cell surface core proteins, while analysis of publicly available RNA-seq datasets indicated expression of *Sdc1*, *Sdc2*, *Sdc4*, *Gpc1*, and *Gpc6* by LLC cells, staining with antibodies directed towards these proteins showed only appreciable expression of Sdc1, Sdc2, and Gpc1 (Supplementary Fig. 5E); thus, we chose to explore the contribution of each of these core proteins using our CRISPR-mediated knockout strategy. Knockouts of secreted ECM (*Hspg2*) and cell surface-localized (*Sdc1*, *Sdc2*, *Gpc1*) HS core proteins were confirmed at the protein level via flow cytometry (Supplementary Fig. 5F). Surprisingly, despite the concept in the HS field that core proteins are generally not a determinative factor in HSBP signaling, we found that in *Gpc1*-KO sublines (but not *Sdc1*-KO, *Sdc2*-KO, or *Hspg2*-KO), Areg signaling was significantly reduced compared to WT controls (Fig. 3E). Strikingly, despite the lack of Gpc1 protein in KO cell lines, these cells showed fully intact presentation of HS on their surface using 10E4 and JM403 antibodies, likely presented by other cell surface-localized HS core proteins (Fig. 3F); thus, this reduction in Areg signaling in *Gpc1*-KO sublines is not the result of global defects in HS presentation. To confirm the specificity of this signaling reduction, we performed rescue experiments using *Gpc1*-KO cells, reinstating expression via transduction with a retroviral vector encoding *Gpc1* (Supplementary Fig. 5G). Upon rescue of *Gpc1* expression, Areg signaling was fully restored (Fig. 3G). From these experiments, we conclude that glypicans (i.e., GPI-anchored HS-presenting core proteins) are critical for proper Areg interaction with HS in a signaling context.

## Glypican-4 is critical for proper Areg signaling in primary Areg-responsive cells and is upregulated on Col14-LMC in the context of viral lung infection

We next sought to determine if specific HS-presenting core proteins (glypicans) also serve as critical determinants of Areg signaling in LMC. We first queried our previously published bulk RNA-seq dataset and a publicly available scRNA-seq dataset to inform which cell surface core proteins are present in mouse LMC[3,5] (Supplementary Fig. 6A, B). Across bulk RNA-seq and scRNA-seq datasets, various syndecans (*Sdc1*, *Sdc2*, *Sdc3*, *Sdc4*) and glypicans (*Gpc1*, *Gpc3*, *Gpc4*, *Gpc6*) show appreciable transcriptional expression; however, at the protein level, Sdc2 was the most highly expressed syndecan and Gpc4 was the only glypican for which we could detect appreciable expression levels (Fig. 4A).

In order to experimentally address the importance of these HS core proteins for Areg signaling in LMC, we utilized an siRNA-mediated knockdown approach (Fig. 4B). Treatment of LMC with *Gpc4*-targeting siRNA significantly reduced Areg signaling as compared to LMC treated with control (non-targeting) siRNA, whereas treatment with *Sdc2*-targeting siRNA did not affect Areg signaling (Fig. 4C).

We then sought to determine the expression levels of glypicans and syndecans on different LMC subsets in vivo. We hypothesized that certain cell surface HS core proteins might be induced by specific LMC subsets in the context of tissue damage. Thus, we stained for Sdc2 and

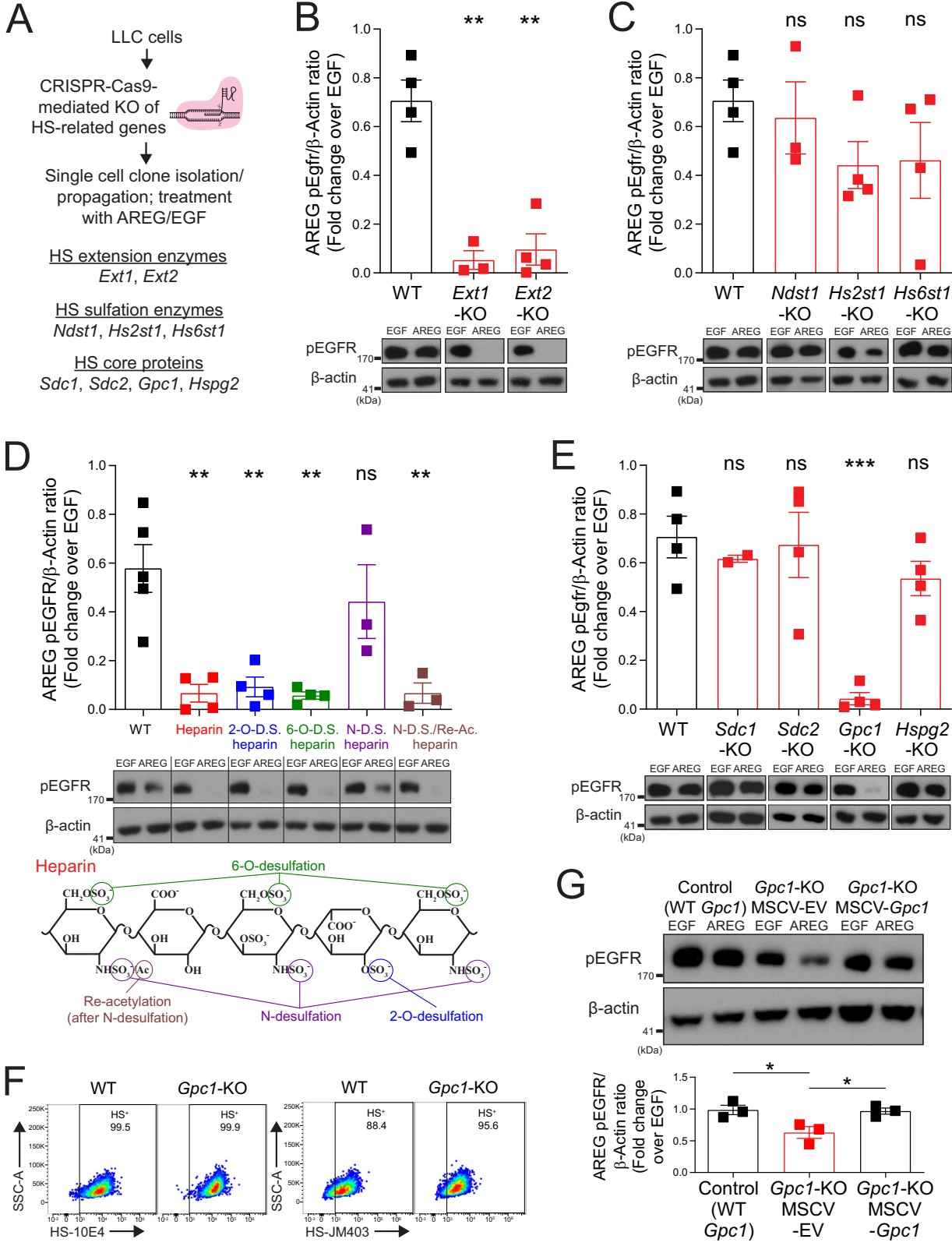

Gpc4 in lungs undergoing models of lung damage. Strikingly, we found that Gpc4 is highly upregulated on Col14-LMC in the context of IAV infection (Fig. 4D); no change from baseline was seen in Col13-LMC, while Hhip-LMC showed a change of a much smaller magnitude. Conversely, Sdc2 did not show similar increases; in fact, it showed significant downregulation on Col14-LMC and Col13-LMC during IAV

infection (Fig. 4E). In the context of bleomycin treatment, Gpc4 was found to be upregulated on all mesenchymal subpopulations (Supplementary Fig. 6C), suggesting that different sources of damage may influence Areg responsivity of particular LMC subsets. These results indicate that Gpc4 upregulation in the context of IAV infection is largely restricted to the Col14-LMC subset.

**Fig. 3 | HS-related gene knockout cell lines identify glypican-1 as a critical HS core protein for Areg signaling. A** Experimental schematic for generation of knockout (KO) panel of HS-related genes by CRISPR-Cas9 in LLC cells. Western blotting for phospho-EGFR (Y1068) and β-actin of WT ($n = 4$) and *Ext1*-KO ($n = 3$) or *Ext2*-KO ($n = 4$) LLC cell sublines (**B**), of WT ($n = 4$) and *Ndst1*-KO ($n = 3$), *Hs2st1*-KO ($n = 4$), or *Hs6st1*-KO ($n = 4$) LLC cell sublines (**C**), or WT ($n = 4$) and *Sdc1*-KO ($n = 2$), *Sdc2*-KO ($n = 4$), *Gpc1*-KO ($n = 4$), or *Hspg2*-KO ($n = 4$) LLC cell sublines (**E**), stimulated (15 min) with mouse AREG (500 ng/ml) or EGF (100 ng/ml). AREG phosphorylation level quantification was done by adjusting to EGF controls. Representative western blots shown. WT control data in (**C**, **E**) are the same as in (**B**), as these experiments were run and analyzed on the same blots. Graph contains all values from three to four experiments (across two clonal sublines per gene, for all except *Sdc1* [only one clone]). **D** Western blotting for phospho-EGFR (Y1068) and β-actin of WT LLC stimulated (15 min.) with mouse AREG (500 ng/ml) or EGF (100 ng/ml) that had been pretreated (15 min) with vehicle ($n = 5$) or heparin (unaltered [$n = 4$] or chemically desulfated at specific sites: 2-O-desulfated [$n = 4$], 6-O-desulfated [$n = 4$],

N-desulfated [$n = 3$], N-desulfated/reacetylated [$n = 3$]). AREG phosphorylation level quantification was done by adjusting to EGF controls. Representative western blots shown. Graph contains all values from three to four experiments. Schematic depicts targeted desulfation/re-acetylation sites in differentially desulfated heparin variants. **F** Flow cytometry using HS-directed antibodies 10E4 and JM403 on WT or *Gpc1*-KO LLC. Representative plots shown from three experiments. **G** Western blotting for phospho-EGFR (Y1068) and β-actin of CRISPR-Cas9 control (WT *Gpc1*), *Gpc1*-KO LLC transduced with empty vector MSCV retrovirus (MSCV-EV), or *Gpc1*-KO LLC transduced with MSCV retrovirus with *Gpc1* mRNA (MSCV-*Gpc1*), stimulated (15 min.) with mouse AREG (500 ng/ml) or EGF (100 ng/ml). AREG phosphorylation level quantification was done by adjusting to EGF controls. Representative western blots shown. $n = 3$ per condition, graph contains all values from three experiments. Statistical analysis done for western blot data using two-tailed unpaired *t*-tests. Mean and standard error displayed on graphs; ns not significant, *$0.01 < p < 0.05$, **$0.001 < p < 0.01$, ***$0.0001 < p < 0.001$. Source data are provided as a Source Data file.

## HS affects Areg-related tissue repair pathways in vivo

To test the necessity of HS for proper Areg-mediated tissue repair in vivo, we conditionally deleted *Ext1* in lung LMC that express *Col1a2* by crossing Col1a2-CreER mice with *Ext1*[fl/fl] mice[32,33]. To assess knockout efficiency in various cell types, we sorted lung cells from tamoxifen (TMX)-treated Col1a2-CreER⁻ *Ext1*[fl/fl] control or Col1a2-CreER⁺ *Ext1*[fl/fl] mice and performed PCR for the targeted genomic region in the *Ext1* gene (Supplementary Fig. 7A). Among broad cell populations, mesenchymal cells show the highest level of *Ext1* deletion compared to hematopoietic, endothelial, and epithelial cells. Among LMC subpopulations, only Col14-LMC shows the maximum level of deletion, in line with our previous studies showing that the *Col1a2* gene is most highly expressed in this subpopulation[3].

To confirm the specificity and efficacy of HS KO we harvested lungs from TMX-administered control (Col1a2-CreER⁻) and Col1a2-CreER⁺ *Ext1*[fl/fl] (HS[cKO]) mice and performed flow cytometry with the 10E4 HS-targeted antibody. Only Col14-LMC shows a significant reduction in HS staining in HS[cKO] mice (Fig. 5A). Interestingly, contrary to the accepted narrative in the field of HS study that tissue cells have ubiquitous expression of HS, we found that staining for HS was in fact fairly low in Col14-LMC as compared to other cell types. Strikingly, when we performed HS staining on lungs of control and HS[cKO] mice undergoing the lung damage models described below (IAV and bleomycin), we found Col14-LMC substantially upregulate levels of HS in the setting of tissue damage (Fig. 5B, Supplementary Fig. 7E). Among broader cell types, this upregulation does not occur for hematopoietic cells, endothelial cells, epithelial cells, or even in total mesenchymal cells; among other LMC subsets, Col13-LMC actually show a significant decrease in HS upon lung damage, and while a significant increase is apparent for Hhip-LMC, it is of lesser magnitude/to a lower maximal level than for Col14-LMC. With regards to HS downregulation in HS[cKO] mice undergoing these models, Col14-LMC shows the greatest loss of HS in HS[cKO] mice compared to control animals. Hematopoietic, endothelial, and epithelial cells show no loss of HS, while significant differences are apparent in total mesenchymal cells. Col13-LMC (during IAV infection only) and Hhip-LMC (during bleomycin treatment only) exhibit significant reductions in surface HS, but to a lesser degree than that seen for Col14-LMC (Fig. 5B, Supplementary Fig. 7E). Thus, we conclude from these data that (1) HS is in fact a damage-inducible modality for Col14-LMC, rather than a ubiquitously expressed mediator, and (2) the Col1a2-CreER driver most effectively targets Col14-LMC among mesenchymal subtypes.

Given that prior work from our group has shown that Treg cell-derived Areg signaling to Col14-LMC is critical for tissue repair following IAV infection[2,3], we sought to ascertain the role of HS in this interaction by comparing TMX-treated control (Col1a2-CreER⁻) and HS[cKO] mice during IAV infection; PBS mock-infected, TMX-treated Col1a2-CreER⁻ mice were also included to establish an untreated

baseline of our readouts (Fig. 5C). Similar to previous studies in Treg cell-specific Areg KO mice[2], we found no changes in weight loss or body temperature between IAV-infected control and HS[cKO] mice (Supplementary Fig. 7B, C). However, at the level of blood oxygen saturation (SpO₂), we found a significant decrease over the course of disease—with maximal differences seen at 7 days post-inoculation (d.p.i.)—in IAV-infected HS[cKO] mice compared to IAV-infected controls (Fig. 5D). We additionally saw a significant decrease in the proliferation of Col14-LMC in IAV-infected HS[cKO] mice compared to IAV-infected control animals at 8 d.p.i. (Fig. 5E), which is likely indicative of a failure to properly activate these cells in the context of HS deficiency. Importantly, we saw no differences in overall viral load (Fig. 5F), AREG production by Treg cells (Supplementary Fig. 7D), or immune cell infiltrate between genotypes (besides slight differences in NK cell and γδ T cells) (Supplementary Fig. 8). Together, these results indicate that the deficiencies seen here in HS[cKO] mice are the result of aberrant tissue repair, phenocopying what we previously observed with Treg cell-specific *Areg* deletion or deletion of *Egfr* on Col14-LMC[2,3].

In order to further explore the consequences of HS loss on downstream repair modalities in HS[cKO] mice, we used additional methods to evaluate tissue health in these experiments. First, we analyzed hematoxylin and eosin (H&E)-stained lungs from matched lobes of control IAV-infected and HS[cKO] IAV-infected mice, quantifying areas of inflammation (Supplementary Fig. 9A). We saw no differences in this parameter, consistent with the largely unchanged immune infiltrate profile (Supplementary Fig. 8). We additionally attempted to analyze airway/alveolar damage from histological stains. The small airways/bronchioles in both control and HS[cKO] IAV-infected lungs appear to contain essentially intact epithelial barriers, with minimal chronic inflammation of airway epithelium and little apparent sloughing of epithelial cells at this time point (Supplementary Fig. 9B). Assessment of alveolar epithelial cells in affected alveolar areas of both control and HS[cKO] IAV-infected lungs was almost completely obscured by the immune infiltrate (Supplementary Fig. 9B); thus, histological analysis of damage in alveolar cell types was not able to be performed in this fashion. We also saw no differences in fibrosis at this time point using Masson's Trichrome staining (Supplementary Fig. 9C).

As an alternative method to assess changes to the lung epithelium in this context, we performed flow cytometry with markers for airway cells (i.e., ciliated, club, and basal cells) vs. alveolar cells (i.e., AT1 and AT2 cells) based on the expression of CD24 and CD104 (integrin β4)[34,35] (Supplementary Fig. 9D). We found that there were significant differences in the amount of live airway cells vs. alveolar cells recovered from control vs. HS[cKO] IAV-infected lungs, reflected by a significant decrease in the proportion of recovered alveolar epithelial cells (Supplementary Fig. 9E), a significant increase in the proportion of recovered airway cells (Supplementary Fig. 9F), and a significant decrease in the ratio of alveolar cell to airway cell counts

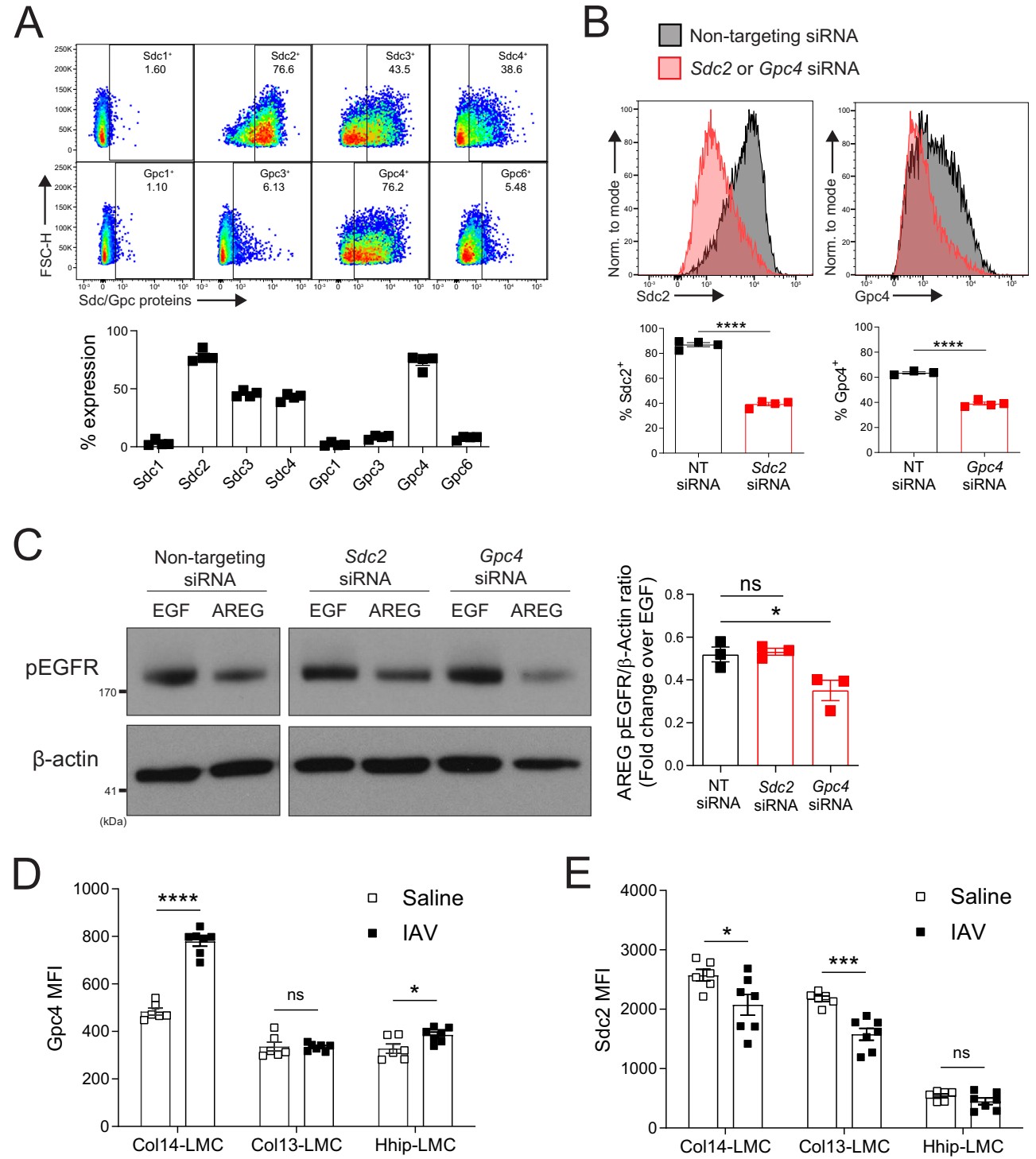

(Supplementary Fig. 9G). Since there appeared to be few alterations in airways based on histology (Supplementary Fig. 9B), these metrics are seemingly an indicator of fewer live alveolar cells recovered from HS[cKO] IAV-infected lungs compared to control IAV-infected lungs, reflecting increased alveolar epithelial damage in this context. When coupled with the observed decreased blood oxygen saturation, as well as the decrease in the proliferation of reparative mesenchymal cells, this provides further evidence for deficient epithelial repair in the HS[cKO] context.

To investigate whether this failure to properly activate tissue repair activity is indeed Areg-dependent, we utilized an antibody-blocking approach that has previously shown efficacy in other Areg-related lung repair models[36] (Fig. 5G). As compared to IgG-treated controls, mice treated with αAreg antibody exhibited reduced blood oxygen saturation following IAV infection (Fig. 5H), mirroring the decrease seen in IAV-infected HS[cKO] mice (Fig. 5D). No further reduction in blood oxygen saturation was observed upon IAV infection of HS[cKO] mice treated with αAreg antibody. These results imply that the defects in tissue repair seen in HS[cKO] mice are indeed Areg-dependent.

To extend our investigations using this mouse line to other forms of lung damage, we utilized the bleomycin model of sterile lung damage, which causes extensive alveolar damage and lung fibrosis. Despite the profound loss of HS seen on Col14-LMC in this model (Supplementary Fig. 7E), disease induction—as indicated by both

**Fig. 4 | Glypican-4 is critical for proper Areg signaling in primary Areg-responsive cells and is upregulated on Col14-LMC in the context of viral lung infection. A** Flow cytometry using antibodies targeting Sdc1, Sdc2, Sdc3, Sdc4, Gpc1, Gpc3, Gpc4, or Gpc6 on cultured bulk LMC (negative bead enrichment for CD45⁻CD31⁻Epcam⁻ cells, plated without sorting [see Supplementary Fig. 2]; non-adherent cells washed away/media changed at 16–18 h, then cultured for 48 h). Gated on CD45⁻CD31⁻Epcam⁻Pdgfra⁺ cells post-culture. Representative flow cyto-metry plots shown. Gating based on IgG controls. Percent staining positive dis-played in plots. *n* = 4 per target, graph contains all values from two experiments. **B** Flow cytometry using antibodies targeting Sdc2 or Gpc4 on cultured bulk LMC, treated with non-targeting (NT) siRNA (*n* = 4 for *Sdc2* siRNA control, *n* = 3 for *Gpc4* siRNA control) or siRNA targeting either *Sdc2* (*n* = 4) or *Gpc4* (*n* = 4) (negative bead enrichment as in **A**; non-adherent cells washed away/media changed after 16–18 h, then treated with 25–50 nM siRNA for 48 h). Gated on CD45⁻CD31⁻Epcam⁻Pdgfra⁺ cells post-culture. Representative flow cytometry plots shown. Graphs contain all values from two to three experiments. **C** Western blotting for phospho-EGFR (Y1068) and β-actin of bulk LMC treated with NT siRNA, *Sdc2* siRNA, or *Gpc4* siRNA

as in B (50 nM, 48 h), then stimulated (15 min) with mouse AREG (500 ng/ml) or EGF (100 ng/ml). AREG phosphorylation level quantification was done by adjusting to EGF phosphorylation controls. Representative western blots shown. *n* = 3 per condition, graph contains all values from three experiments. **D** Gpc4 protein expression determined by flow cytometry (MFI median fluorescence intensity), using a Gpc4-directed antibody, on freshly harvested LMC subsets, from either saline-treated (*n* = 6) or IAV-infected (*n* = 7) (300 TCID50) lungs (8 d.p.i.). Gating strategy in Supplementary Fig. 2 (total lung). Graph contains all values from two experiments. **E** Sdc2 protein expression determined by flow cytometry (MFI median fluorescence intensity), using a Sdc2-directed antibody, on freshly harvested LMC subsets, from either saline-treated (*n* = 6) or IAV-infected (*n* = 7) (300 TCID50) lungs (8 d.p.i.). Gating strategy in Supplementary Fig. 2 (total lung). Graph contains all values from two experiments. Statistical analysis done for wes-tern blot/flow cytometry data using two-tailed unpaired *t*-tests. Mean and standard error displayed on graphs; ns not significant, *0.01 < *p* < 0.05, ***0.0001 < *p* < 0.001, ****p* < 0.0001. Source data are provided as a Source Data file.

weight loss (Supplementary Fig. 7F) and staining for α-smooth muscle actin (α-SMA) on LMC as a proxy for fibrosis (Supplementary Fig. 7G)—was not significantly altered.

### HS on Col14-LMCs confers proper responsiveness to Treg cell-derived signals

Finally, to explore the impact of Col14-LMC HS on Treg cell-mediated signaling, we utilized a Col14-LMC/Treg cell co-culture system wherein Col14-LMC from control and HS^cKO mice were sorted and co-cultured with primary Treg cells from IAV-infected lungs (8 d.p.i.) (Fig. 6A). Staining for HS in Col14-LMC showed an essentially complete loss of HS on cells from HS^cKO mice compared to controls (Fig. 6B). Tran-scription of *Lif*, previously described to be Treg cell inducible[3], was significantly reduced in HS^cKO Col14-LMC following incubation with Treg cells, compared to control Col14-LMC (Fig. 6C). This reduction was also observed for *Il6*, for which expression has previously been shown to be Treg cell-inducible in astrocytes[37] (Fig. 6C). Following incubation with Treg cells, angiogenesis mediator *Vegfa* showed a trend toward increased expression in control Col14-LMC, while no increase was seen in HS^cKO cells expression (Fig. 6C). Notably, for all genes analyzed here, baseline (without incubation with Treg cells) decreases in transcription were observed for HS^cKO Col14-LMC vs. control Col14-LMC (Fig. 6C); this may be indicative of a baseline defi-ciency in the tissue repair capabilities of Col14-LMC that lack HS.

To test whether these phenotypes were dependent on Treg cell-derived Areg, αAreg or IgG control antibodies were included in co-cultures. In this setting, we found that *Lif* expression was significantly decreased in the presence of αAreg antibody for both control and HS^cKO Col14-LMC, but to a lesser magnitude in the latter (Fig. 6D). Additionally, *Vegfa* gene expression was significantly reduced upon the addition of αAreg in control Col14-LMC co-cultures, while in co-cultures with HS^cKO Col14-LMC it was unaltered (Fig. 6D). *Il6* did not show alteration in either control or HS^cKO Col14-LMC in this context (Fig. 6D). Ultimately, these experiments highlight a reduction in the responsiveness of Col14-LMC to Treg cells (in a partially Areg-dependent manner) and alterations in the baseline transcriptional state of Col14-LMC when these cells lack HS, which likely underlies deficiencies in their reparative capacity.

### Discussion

In this report, we sought to gain a deeper understanding of the manner that Areg, an immune system-derived growth factor involved in tissue repair during damage, interacts with HS. In doing so, we made several discoveries that serve to expand our understanding of the biology of Areg–HS interactions in the context of repair from lung damage.

Previous reports on HSBPs generally regard HS on cognate cells as an obligate member of a signaling complex involving a receptor, an HS-binding ligand, and HS itself. For example, the FGF family is known to

require the formation of a ternary complex between itself, HS, and cognate FGFRs for signaling[38–40]. In light of this and previous work on the HS-binding capability of Areg[12,13], we expected that Areg signaling would be absent in cells where HS has been disrupted. However, by testing for downstream MAPK and AKT-mTOR activation in various HS-inhibition scenarios, including a completely HS-deficient cell line, we reveal that these pathways maintain activity in response to Areg even in the absence of proper HS engagement, pointing to the existence of HS-dependent and HS-independent signaling modalities for Areg. We used this finding as an opportunity to explore the transcriptional consequences of this previously unappreciated dichotomy with RNA-seq on an Areg-responsive, reparative mesenchymal cell subpopula-tion from the lung (Col14-LMC). Strikingly, we found that certain Areg-induced transcriptional pathways in these cells are HS-independent (such as expression of tissue factors, growth factors, and angiogenesis mediators), while others are HS-dependent (such as the Hippo path-way, actin mobilization, intermediate filament induction, and gap junction formation). We also found that the expression of certain FGF and Wnt reparative mediators share this HS-dependent pattern. Based on the in vivo data we present herein, we believe that the HS-dependent expression modalities we have identified are critical in mediating proper reparative function in Areg target cells. In support of this, activation of the Hippo pathway effector YAP has been shown to promote tissue regeneration in the context of skin wound healing[41]. Areg's HS-binding domain has also been shown to mediate its locali-zation to cell–cell contacts, which could be related to the ability of cells to alter their orientation, polarity, and communication with other cells by engaging these pathways[42]. Furthermore, the biological implica-tions for the existence of HS-independent and -dependent signaling modalities are widespread; the presence or absence of HS at a given signaling interface could plausibly lead to different outcomes for any HSBP.

Another debate in the HS field addressed here involves whether the HS-binding ability of a given HSBP is determined by the core protein on which HS is displayed or the sulfation pattern present in HS. In this regard, cell surface HS core proteins are generally thought to be exchangeable, passive displayers of HS, with differences in binding by varying HSBPs conferred instead by the sulfation pattern present on the displayed HS (as determined by the unique set of sulfation enzymes expressed by a given cell type)[9]. In support of sulfation motifs determining the binding of a given HSBP, certain HSBPs have been shown to bind HS sequences with distinct sulfation patterns, for instance in the widely studied antithrombin–HS interaction[43]. Thus, we expected that the knockout of specific sulfation enzymes would lead to the most significant reduction in Areg signaling potential. Contrary to this paradigm, we found that deletion of a specific GPI-anchored HS core protein (Gpc1) showed the greatest reduction in Areg signaling, while knockout of sulfation enzymes showed little effect. Importantly,

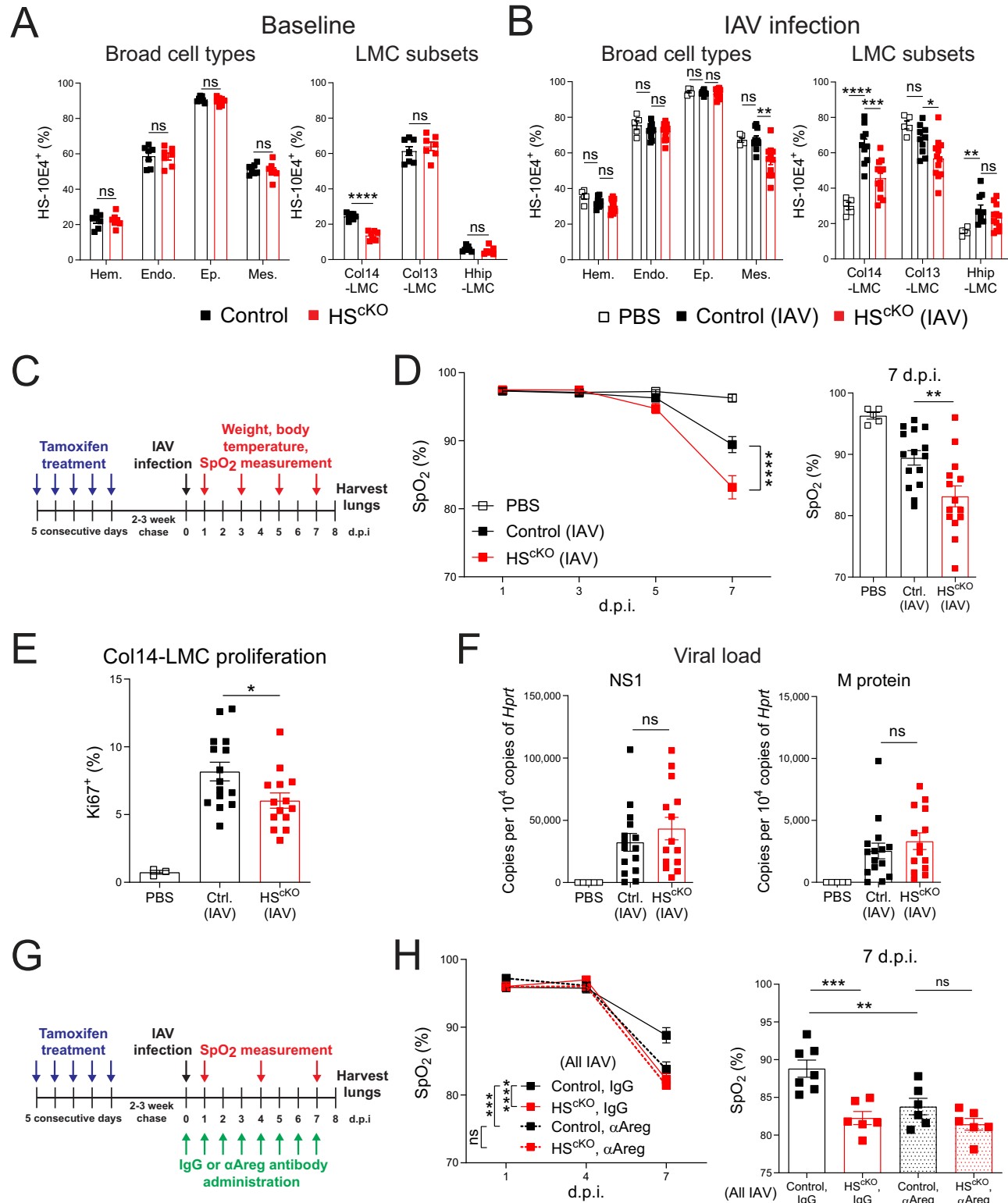

knockout of other cell surface HS core proteins did not mediate the same reduction in Areg signaling as Gpc1; furthermore, in Gpc1 knockout cells, a full complement of HS was still displayed on the cell surface, presumably by other core proteins. This result was validated in primary mouse LMC upon siRNA-mediated knockdown of Gpc4, the only glypican expressed by this cell type. Pre-binding experiments in which Areg was exposed to heparin of different sulfation states prior to stimulation of cells further implicate Areg as a generalist, sulfation-agnostic HS binder, showing no heightened affinity for specific

sulfation sites. These findings suggest that for each HSBP, careful systematic investigation is necessary to determine the nature of HS-binding ability.

Interestingly, glypicans (due to their GPI-linked nature) are localized to lipid rafts in cell membranes[44]. A previous study found that this localization has the effect of sequestering FGF2 (an HSBP) away from its non-lipid raft-localized receptor, thus inhibiting its signaling[44]. However, EGFR is known to preferentially localize to lipid rafts[45], which would theoretically confine it and glypicans to a similar location on the

**Fig. 5 | HS affects Areg-related tissue repair pathways in vivo.** HS presence, assessed by flow cytometric staining for the HS-directed 10E4 antibody, on indicated cell populations from control ($n = 7$) and HS$^{cKO}$ ($n = 7$) mouse lungs at baseline (all TMX-treated) (**A**), or from mock-infection control (PBS) mice ($n = 5$), IAV-infected (100 TCID50) control mice ($n = 10$), and IAV-infected HS$^{cKO}$ mice ($n = 12$) at 8 d.p.i. (all TMX-treated) (**B**). Hem. hematopoietic, Endo. endothelial, Ep. epithelial, Mes. mesenchymal. The gating strategy in Supplementary Fig. 2 (total lung). Gating based on FMO controls. Graphs contain all values from three experiments. **C** Experimental schematic of IAV model used in (**D**–**F**). **D** Blood oxygen saturation (SpO$_2$) levels in mock-infection control (PBS) mice ($n = 5$), IAV-infected (100 TCID50) control mice ($n = 15$), and IAV-infected HS$^{cKO}$ mice ($n = 14$) (all TMX-treated), with the same mice undergoing repeated measurement at indicated time points. Right: All SpO$_2$ values from 7 d.p.i. Graphs contain all values from three experiments. **E** Proliferation induction, assessed via flow cytometry using a Ki67-directed antibody, in Col14-LMC from lungs of mice in (**D**). Gating strategy for Col14-LMC in Supplementary Fig. 2 (total lung). Gating based on FMO control. The graph contains all values from three experiments. **F** qPCR of IAV viral NS1 and M protein in lungs from mice in (**D**, **E**). Expression values computed as copies of target gene per 10,000 copies of the housekeeping gene (*Hprt*). Graphs contain all values from three experiments. **G** Experimental schematic of IAV model used in (**H**); daily αAreg or normal goat IgG (control) administration done with 5 μg in 200 μl PBS i.p. **H** SpO$_2$ levels in IAV-infected control mice treated with IgG control antibody ($n = 7$) or anti-Areg antibody ($n = 6$) and HS$^{cKO}$ mice treated with IgG control antibody ($n = 6$) or anti-Areg antibody ($n = 6$) (all TMX-treated), with the same mice undergoing repeated measurement at indicated time points. Right: All SpO$_2$ values from 7 d.p.i. Graphs contain all values from three experiments. Statistical analysis done for SpO$_2$ data across all time points using two-way repeated measures ANOVA and for flow cytometry/qPCR/individual time point SpO$_2$ data using two-tailed unpaired *t*-tests. Mean and standard error displayed on graphs; ns not significant, *$0.01 < p < 0.05$, **$0.001 < p < 0.01$, ***$0.0001 < p < 0.001$, ****$p < 0.0001$. Source data are provided as a Source Data file.

cell membrane and potentiate signaling. While plausibly supported by our results, high-magnification imaging studies to observe subcellular localization of glypicans and EGFR would be necessary to confirm this interaction. Moreover, in vivo models of specific glypican knockouts, which we did not attempt in this work, could be utilized to explore their roles in Areg/EGFR-mediated tissue repair. If further research is done in this regard, it could provide fresh insight into the biology of EGFR signaling, one of the most well-studied pathways in biology.

One additional paradigm related to HS biology addressed in our work is the assumption that tissue parenchymal cells ubiquitously express HS at baseline[11]. While constitutively high expression levels were apparent on the cancer cell lines used in this study, for tissue cells from uninfected mouse lungs, we found that HS levels were in fact quite variable, including relatively low expression on Col14-LMC, a subpopulation known to be critical for tissue repair. Surprisingly, we found that HS is highly upregulated on Col14-LMC (but not altered or only slightly upregulated on other cell types) during lung damage. Furthermore, Gpc4, the HS core protein we identified as preferentially mediating Areg signaling in LMC, similarly shows specific upregulation on Col14-LMC compared to other LMC subsets during IAV infection. The idea of HS as a damage-induced mediator of tissue repair on certain cell subsets runs counter to the concept of ubiquitous expression of HS on tissue cell populations and suggests potential future research directions wherein it may also be found to alter growth factor signaling in disease-specific contexts. The intracellular signaling pathways leading to HS and Gpc4 upregulation merit further investigation; this could plausibly result from inflammatory signaling within the lung environment, sensing of mediators from damaged epithelium, or direct damage to these cells.

In our in vivo investigations, we found that mice with an inducible mesenchymal-specific deletion for HS exhibited a reduced ability to recover blood oxygen saturation following IAV exposure, which we mechanistically attributed to improper Areg signaling in HS-deleted Col14-LMC. Notably, there were no differences in weight loss, body temperature, immune infiltrate, AREG production by Treg cells, or viral load (IAV uses sialic acid, not HS for viral entry[46]). The observed decrease in blood oxygen saturation, coupled with the normalcy of these other factors, supports the concept that the former is the result of deficient tissue repair in this context. To the best of our knowledge, this is the only known report of mesenchymal HS serving a critical role in tissue recovery during viral lung infection. Beyond boosting our biological understanding of tissue responses to and recovery from viral infection in the lung, this research suggests several potential avenues of investigation into whether similar dependency on mesenchymal HS is at play for tissue repair in other organs, in the context of other tissue-damaging agents, and/or for other HSBPs.

While the experiments in this report largely speak to basic aspects of HS–immune system interaction, there are several possible future clinical applications for our findings. The discovery that HS is critical for Areg-mediated tissue reparative signaling in the context of lung damage may point to the possibility of treatments that could upregulate or stabilize the presence of HS during tissue injury. However, this becomes complicated in the case of viral infection, where many human viruses rely on HS for entry, and thus HS upregulation could increase viral infection levels. Relatedly, while deploying HS or heparin as decoy viral receptors has been proposed as a way to inhibit viral spreading in patients[47,48], their delivery could also have the effect of binding up reparative HSBPs such as Areg during tissue recovery; thus, further work is needed to investigate the ideal temporal deployment of these molecules to curtail viral infection while ensuring proper tissue repair.

In conclusion, we have attempted here to provide experimental insight into HS biology as it relates to the immune-derived tissue mediator Areg, which we believe to be an understudied component of immune cell–non-immune cell interaction during tissue repair. Our findings position Areg as an unconventional HSBP that defies many paradigms held in the glycobiology field, while also highlighting that HS along with certain core proteins can serve as damage-inducible tissue modalities in a way that affects tissue repair mechanisms.

## Methods

### Mice

Animal experiments were approved by Columbia University's Institutional Animal Care and Use Committee (protocol AC-AABT2656). Mice were housed in a facility using a 12 h light/dark cycle, with ambient temperature 20–26 °C, and humidity maintained between 30% and 70%. All mice used for experiments were of the *Mus musculus* species. All mice used for experiments were 8–12 weeks old. Euthanasia was performed by carbon dioxide asphyxiation, and secondarily by cervical dislocation. Wild-type (WT) mice (background: C57BL/6N) were acquired and bred from Jackson Laboratory stocks (Strain #:005304). These mice or lab-bred descendants were utilized for bulk LMC isolation, Col14-LMC isolation/sorting, and in vivo/freshly harvested analysis of LMC populations. Col1a2-CreER mice were acquired from Jackson Laboratory (Strain #:029567) (background: C57BL/6J), and were previously described[32]. *Ext1*$^{fl/fl}$ mice were a generous gift from the laboratory of Dr. Yu Yamaguchi (Sanford Burnham Prebys) (background: C57BL/6J) and were previously described[33]. Col1a2-CreER mice were crossed to *Ext1*$^{fl/fl}$ mice; these mice were bred with Col1a2-CreER in a hemizygous fashion (Col1a2-CreER$^+$ as referred to in this report), with *Ext1*$^{fl/fl}$ as homozygous. These mice showed no overt physical differences, induction of weight loss, or symptoms of distress either at baseline or following tamoxifen treatment. Col1a2-CreER$^+$ Ext1$^{fl/fl}$ mice were tested for leaky deletion in *Ext1* (i.e., without tamoxifen induction), which was shown not to occur over multiple litters. Mice were screened for the C57BL/6N WT *Nnt* allele (as opposed to the mutant

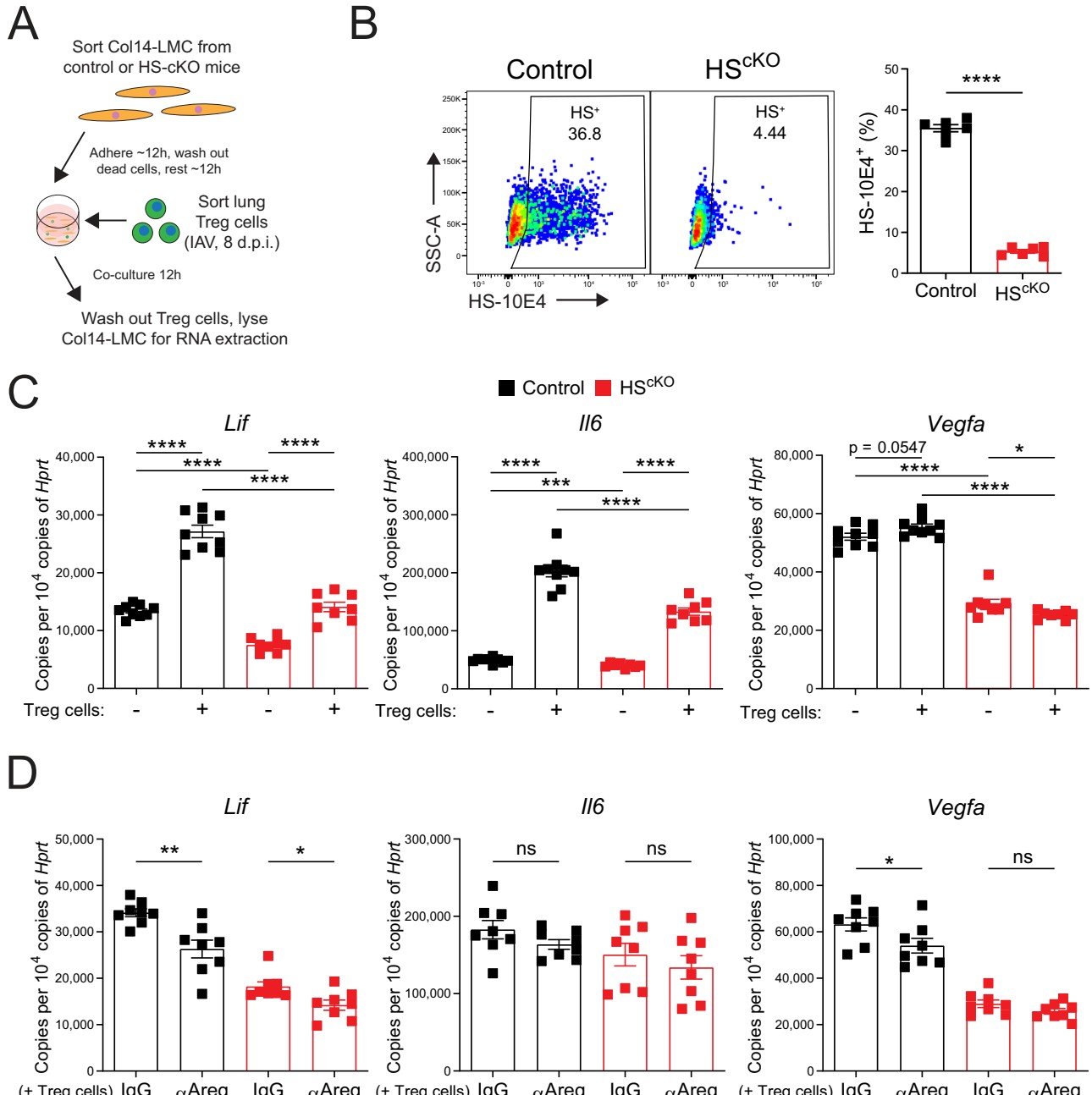

**Fig. 6 | HS on Col14-LMCs confers proper responsiveness to Treg cell-derived signals. A** Experimental schematic for Col14-LMC/IAV-induced lung Treg cell co-culture experiments. **B** HS presence, assessed by flow cytometric staining for the HS-directed 10E4 antibody, on Col14-LMC sorted from control (*n* = 6) and HS^cKO (*n* = 6) mouse lungs (baseline) and cultured for 48 h (non-adherent cells washed out and media replaced at 12 h), with the gating/sorting strategy described in Supplementary Fig. 2 (negative bead enriched). Representative flow cytometry plots shown. Gating based on FMO control. Percent staining positive displayed in plots. The graph contains all values from two experiments. **C** qPCR for IAV (300 TCID50) lung Treg cell-inducible genes *Lif*, *Il6*, and *Vegfa* from control (*n* = 9) and HS^cKO (*n* = 8) Col14-LMC without inclusion of IAV-induced lung Treg cells, or control (*n* = 9) and HS^cKO (*n* = 8) Col14-LMC with inclusion of IAV-induced lung Treg cells. Expression values computed as copies of target gene per 10,000 copies of the housekeeping gene (*Hprt*). Expression values were normalized across experiments. Graphs contain all values from three experiments. **D** qPCR for Treg cell-inducible genes *Lif*, *Il6*, and *Vegfa* from control (*n* = 8) and HS^cKO (*n* = 8) Col14-LMC co-cultured with IAV-induced lung Treg cells and control IgG antibody (2 μg/ml), or control (*n* = 8) and HS^cKO (*n* = 8) Col14-LMC co-cultured with IAV-induced lung Treg cells and αAreg antibody (2 μg/ml). Expression values computed as copies of target gene per 10,000 copies of the housekeeping gene (*Hprt*). Expression values were normalized across experiments. Graphs contain all values from three experiments. Statistical analysis done for flow cytometry/qPCR data using two-tailed unpaired *t*-tests. Mean and standard error displayed on graphs; ns not significant, *0.01 < *p* < 0.05, **0.001 < *p* < 0.01, ***0.0001 < *p* < 0.001, ****p* < 0.0001. Source data are provided as a Source Data file.

*Nnt*^C57BL/6J allele), with all mice used for experiments containing at least one copy of the WT allele. *Foxp3*^EGFP mice were a generous gift from the laboratory of Dr. Alexander Rudensky (Memorial Sloan Kettering) (background: C57BL/6J), and were previously described[49].

**Cell lines**

LLC cells (also known as LL/2 or LLC1) were obtained from the American Type Culture Collection (ATCC, Catalog # CRL-1642). Phoenix-ECO cells were obtained from the American Type Culture Collection

(ATCC, Catalog # CRL-3214). A549 cells were a generous gift from the laboratory of Dr. Richard Vallee (Columbia University). Ba/F3 cells were a generous gift from the laboratory of Dr. Michael Green (UMass Chan Medical School). LLC cells, A549 cells, and Phoenix-ECO cells were cultured in DMEM with 100x penicillin/streptomycin (Gibco), 100x GlutaMAX (Gibco), and 10% fetal bovine serum (FBS) (Corning); cells were cultured on tissue culture-treated plates (Corning). Ba/F3 cells were cultured in RPMI with 100x penicillin/streptomycin (Gibco), 100x GlutaMAX (Gibco), and 10% FBS (Corning); cells were cultured on non-tissue culture-treated plates. For maintenance of Ba/F3 cells, rmIL-3 (Biolegend) was added to media (1 ng/ml). Details for transfection/transduction of cell lines with various methods are outlined in the Supplementary Methods.

## RNA-seq

Col14-LMC were sorted as described (see Supplementary Methods), then plated at 50,000 cells/well in 48 well tissue culture plates. After ~24 h in culture, media was aspirated, dead cells were washed away with 1 wash of DMEM (Gibco), then media was replaced, with either sodium chlorate (3 mg/ml) or vehicle (water) added. After overnight incubation (16–18 h), cells were treated with vehicle (PBS + 1% BSA) or rmAreg (200, 500, or 1000 ng/ml) (R&D Systems). After 4 h of incubation, media was aspirated and cells were lysed in Trizol Reagent (Thermo). RNA was extracted by the Columbia Molecular Pathology Shared Resource using the miRNeasy Micro Kit (Qiagen, Catalog # 217084), and RINs were found to be >9.6 via Bioanalyzer analysis. We worked with the Columbia Genome Center to perform RNA-seq, using poly-A pulldown to isolate RNA, with subsequent library preparation with a Nextera XT Kit (Illumina, Catalog # FC-131-1024) and sequencing with a NovaSeq 6000 sequencer (Illumina) at 40 million reads. Base calling was done with RTA (Illumina) and bclfastq2 (version 2.19) was used to convert BCL to fastq files with adapter trimming. Pseudoalignment was done with kallisto (version 0.44.0) from transcriptomic data (Ensembl v96, Mouse GRCm38.p6). Subsequent analysis including differential expression was done using the DESeq2 package in R, implemented with iDEP (version 1.13)[50]. Pathway analysis was done with g:Profiler (version: database built on 2022-12-28)[51] to analyze representation within KEGG pathways. Secondary data analysis was done on published RNA-seq datasets (NCBI GEO GSE99714[5], GSE103548[31], GSE169127[3]). Seurat (version 3.0) was used to analyze single-cell RNA-seq data[52].

## Mouse tamoxifen treatment, lung damage model induction, and disease assessment

For tamoxifen-based induction of CreER, tamoxifen (Sigma Aldrich) was diluted in corn oil (Sigma Aldrich) at 20 mg/ml, and shaken overnight at 37 °C/250 r.p.m. for dissolution in solution, prior to storage at 4 °C (<1 month). Mice were treated for 5 consecutive days with 100 mg/kg tamoxifen solution (100–150 µl per mouse); following the final tamoxifen treatment, mice were given a 2–3 week chase period prior to experimentation. Influenza A (IAV) virus (PR8/H1N1) was a generous gift from the laboratory of Dr. Donna Farber (Columbia University). For IAV infection, mice were given ketamine/xylazine for anesthesia, then infected intranasally with 100–300 TCID50 of virus diluted in 1x PBS (as determined by the Farber lab using Madin–Darby canine kidney epithelial cell infection assays)[53]. Both male and female mice were used for IAV experiments, and mouse lungs were harvested at 8 days post-inoculation (d.p.i.). Bleomycin (Meitheal Pharmaceuticals) was diluted in sterile 0.9% saline (0.5 U/ml), and 50 µl per mouse was administered via oropharyngeal aspiration (~1 U/kg), following a previously described technique[54]. Briefly, mice were given ketamine/xylazine for anesthesia, then placed on an apparatus suspending them at a 60° angle from horizontal by surgical suture string from their teeth. The tongue was removed from the mouth and held with padded forceps,

then the bleomycin solution was pipetted into the back of the mouth, followed immediately by plugging the nose with padded forceps to induce inhalation of the solution. Male mice were used for bleomycin experiments, and mouse lungs were harvested at 14 d.p.i. Littermate, age-matched mice were used for all experiments. For IAV and bleomycin experiments, mice were weighed every 1–3 days. For IAV experiments, mouse body temperature was assessed with a rectal thermometer. For IAV experiments, mouse blood oxygen saturation (SpO$_2$) was assessed using a MouseOx Plus Pulse Oximeter (Starr Life Sciences). The area around the mouse's neck was shaved and Nair Hair Remover Product was applied to chemically remove residual hair at the time of IAV infection. A Small Mouse Collar Sensor (Starr Life Sciences) was used to assess SpO$_2$ on unanesthetized at indicated time points during IAV infection progression, with mice placed in a 1 L beaker to restrict movement; assessment was taken for 2–5 min per mouse at each time point, with only high-quality, error-free readings taken and averaged to determine a composite SpO$_2$ value. For IAV experiments, mice were excluded from the final analysis if they experienced loss of SpO$_2$ to a level <85% at 3–4 d.p.i. (as this is an indication of lung damage from technical issues with administration, not from IAV infection), if they did not experience a loss of body temperature <38 °C at 7 d.p.i. (as this is evidence of administration issues leading to an unproductive infection), or if there was visible failure to uptake IAV intranasally during infection. At the endpoint of disease models, lung tissue was perfused and (for certain experiments) bronchoalveolar lavage was performed as described in "Lung processing" in Supplementary Methods. For IAV experiments, left lobes were fixed in 10% neutral buffered formalin (Epredia) for histology, middle and inferior lobes were flash frozen for RNA extraction to assess the viral load, and superior and postcaval lobes were processed for flow cytometry. Lobes for histology were sent to Histology Consultation Services for paraffin embedding, sectioning at 5 µm, and H&E staining, then imaged using an Aperio AT2 (Leica) full slide scanner and analyzed using QuPath (version 0.3.0) software for percentage of the inflamed area; analysis was done in a blinded manner. The same paraffin-embedded lobes were again sectioned at 5 µm and additionally stained for Masson's trichrome and imaged, by Columbia University's Molecular Pathology Shared Resource (MPSR). Lobes for RNA extraction were added to Trizol (Thermo), a ¼" Ceramic Sphere (MP Biomedicals) was added to tubes, and tissue was homogenized in a FastPrep-24 (MP Biomedicals) for lysis, followed by RNA analysis (see "RNA extraction and qPCR" in Supplementary Methods). For bleomycin experiments, full lungs were processed for flow cytometry.

## Western blotting, RNA extraction/qPCR, lung processing, negative bead enrichment/sorting, flow cytometry, cloning/transfection/transduction, siRNA treatment, Col14-LMC/Treg cell co-culture

Details for these methods are outlined in the Supplementary Methods. For qPCR analysis in certain experiments, results were calculated as copies of the gene of interest per copies of the housekeeping gene (*Hprt*) using a previously described method[55].

## Statistics and data analysis

GraphPad Prism (v10.1.2) was used for all statistical analyses and graphing. For western blots, flow cytometry, SpO$_2$, or qPCR analysis where two groups were compared, two-tailed unpaired Student's *t*-tests were used. For western blot analysis where three or more groups were compared, one-way ANOVA was used. For post hoc RNA-seq analysis of individual genes where three or more groups were compared, the Kruskal–Wallis test was used. For longitudinal weight loss, body temperature, or SpO$_2$ analysis, two-way repeated measure ANOVA was used. Statistical significance was determined at $p < 0.05$, with further levels of significance reported in figure legends. Sample

size estimation was determined based on previous studies. FlowJo (version 10.7.1), SnapGene Viewer (version 6.0.4), Adobe Illustrator (version 24.1.2), ImageJ (version 1.53m), MacVector (version 17.0), R (version 4.2.2), Microsoft Excel (version 16.0), and Microsoft Power-point (version 16.0) were used to set up experiments, analyze data, and prepare data.

## Reporting summary
Further information on research design is available in the Nature Portfolio Reporting Summary linked to this article.

## Data availability
The RNA-seq data generated in this study have been deposited in NCBI's Gene Expression Omnibus under GEO accession number GSE263616. All other data generated in this study are provided in the Supplementary Information/Source Data file. Source data are provided with this paper.

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

## Acknowledgements

We thank several Columbia University Core Facilities for their contributions to this work: the Microbiology & Immunology Shared Resources, Columbia Stem Cell Initiative (CSCI) Flow Cytometry, the Columbia Genome Center, and the Herbert Irving Cancer Comprehensive Cancer Center Molecular Pathology Shared Resource (MPSR), funded in part through the NIH/NCI Cancer Center Support Grant P30CA013696 (awarded to N.A.). We thank Dr. Alexander Rudensky for contribution of *Foxp3*[EGFP] mice. We thank Dr. Yu Yamaguchi for contribution of *Ext1*[fl/fl] mice. We thank Dr. Donna Farber for contribution of the influenza A virus (PR8/H1N1). We thank Dr. Richard Vallee for contribution of A549 cells. We thank Dr. Michael Green for contribution of Ba/F3 cells. We thank all other Arpaia lab members, as well as S. Iketani and M. Edwards, for their feedback and comments on this work. This work was supported by NIH grants R21AI149657 and R01HL148718 (awarded to N.A.).

## Author contributions

L.F.L. and N.A. designed research; L.F.L., A.K., O.R.R., F.L., and K.d.l.S.-A. performed research; L.F.L., A.K., and A.S. analyzed data; L.F.L. and N.A. wrote the paper.

## Competing interests

A.S. reports grants or contracts from Boehringer Ingelheim and Genentech to institution, consulting fees from Genentech, Gilead, Abbvie, and Veracyte, payment or honoraria from Medscape, Physician's Education Resource, Memorial Sloan Kettering, Bronx Lebanon, and New York University, support for attending meeting or traveling from International Association for Study of Lung Cancer, College of American Pathologists, patents of Device for cell blocks and Major Pathological Calculator Tool, and stocks of Link Biosystems. The other authors claim no competing interests.
