## [Transparent Peer Review file · Nature Communications]

Heparan sulfate regulates amphiregulin programming of tissue reparative lung mesenchymal cells during influenza A virus infection in mice

Corresponding Author: Dr Nicholas Arpaia

Version 0:

Reviewer comments:

Reviewer #1

(Remarks to the Author)

The following points should be addressed:

1. The title is quite broad and doesn't reflect the new and exciting new data. Authors might consider a revision.
2. At end of first paragraph, line 111, authors might consider a summary statement or two highlighting the results presented above and emphasizing what is new and what is confirmatory.
3. An enormous amount of data and presented and care needs to be taken to be sure adequate controls are well described.

Reviewer #2

(Remarks to the Author)

The work presented by Loffredo et al. demonstrates that heparan sulphate expressed on lung mesenchymal cell subpopulations plays important beneficial roles in regulating signalling responses to Treg cell secreted amphiregulin during influenza A infection.

The manuscript is well written and follows a sound and logical progression of experiments that comprehensively demonstrate the importance of heparan sulphate. In my opinion, the experiments and data produced are sufficient to support the conclusions.

To further strengthen Figure 5/S5/S6 experimental evidence and drawn conclusions, it would be beneficial to see the histopathology and immunochemical staining of the harvested lung tissues to get an idea of the extent of expression and localisation of the evaluated cell types/HS, and the consequences on lung tissue inflammation/structural deterioration and fibrosis after viral load and bleomycin treatments. If there was a reason such figure panels weren't presented, then can the authors please comment on this. I note that the description given in the methods seems to suggest that the entire lungs were digested for cell analyses.

Would it be possible to reference the studies of the Farber lab regarding IAV infection and KEC infection assays?

Some minor formatting issues that were detected during the review, although that is not to say all typos/errors were identified. Please thoroughly check the manuscript as necessary.

1. In abstract line 20, heparan sulphate is written in full despite that previously the abbreviation was given. The full name of IAV, which includes "virus" is not given in abstract line 22.
2. Fig. 1B is called out twice, in two separate paragraphs. It is suggested that these two instances are combined into one paragraph for ease of reading.
3. Line 299, switch order of "maximum" and "the."
4. Line 353, additional "in" can be deleted.

Reviewer #3

(Remarks to the Author)

In the paper "Heparan sulfate regulates amphiregulin signaling towards reparative lung mesenchymal cells during influenza A infection" from Loffredo et al., the authors described heparan sulfate (HS) as new regulator of AREG signaling in Col14-LMC in vitro and during IAV infection in mice. They concluded that Treg-derived AREG induced reparative signaling in Col14-LMC during IAV infection in a HS-dependent manner.

The study is well done with appropriate methods and controls. The paper is well written and the data are presented in an understandable way. It is a follow-up study based on previous publications of the group. Here, the authors described a novel mechanism by which the effect of AREG on lung tissue cells can be regulated during lung infection.

However, I have some major concerns regarding the conclusion, which I think has not been properly shown. The only repair parameter the authors analyzed after IAV infection was Ki-67 in isolated Col-14-LMC. No data from lung tissue were shown, neither data that would allow a correlation of Ki-67 expression with AREG or Treg-derived AREG in the mouse model.

Major points:

Figure 5: The authors showed a reduced proliferation of Col14-LMC after viral infection in HScKO mice. Did they also observe altered AREG/HS downstream signaling in Col14-LMC from these mice?

The authors showed SpO₂ as only disease parameter. If the lung tissue is damaged, they should show this in histological stainings. Was the lung tissue damage altered in HScKO compared to control mice? Did Col-14-LMC express Ki-67 in lung tissue of HScKO mice and was Ki-67 expression localized to areas of tissue damage/repair?

Where AREG levels altered in HScKO compared to control mice? Did Tregs express AREG in HScKO mice and localized near Col-14-LMC?

Was Ki-67 expression of Col14-LMC altered after blockage of AREG? Did the mice develop more severe lung tissue damage?

Why did bleomycin-treated HScKO mice show no differences in SpO₂ compared to control mice? The authors should at least discuss this finding.

Figure 6: Protein expression analysis of Il6 and Vegfa would strengthen the gene expression data. It is not clear why these three genes were analyzed. Are they involved in regenerative/repair processes of Col14-LMC in lung disease? Is expression of Ki-67 induced/altered in this experimental setup?

Minor points:

Fig. 1B: The authors showed three values per group and in the figure legend they stated "n=3 per condition, graphs contain all values from 3 separate experiments". I think that the mean values of three experiments are depicted. However, in Fig. 1C, they showed 3-4 values derived from two experiments. This needs to be clarified.

The authors used a high concentration of murine AREG (500 ng/ml) in vitro to stimulate the cells. I wonder if this high concentration is necessary to see an effect of AREG, how can these data be transferred to the in vivo situation where Tregs (always low in number) will never produce such high AREG levels?

Figure 2: Why is it necessary to include nearly the same number of DEGs in vehicle-treated Col14-LMC and NaClO₃-treated Col14-LMC for KEGG analysis? To achieve this, the authors compared Col14-LMC that were cultured with different AREG concentrations (200 ng/ml vs 1000 ng/ml). It is known that the effect of AREG on other cells depends on the concentration used for the in vitro experiments. Are the results the same, if Col14-LMC treated with the same AREG concentration are compared?

In Fig. 4, the authors used the term bleomycin infection. I think this is not correct because bleomycin is a substance and not a virus. In the Figure legend of S5C, the authors also wrote bleomycin-treated (14 d p.i.). If p.i. stands for post infection, this is also not correct.

Reviewer #4

(Remarks to the Author)

Reviewer #5

(Remarks to the Author)

I co-reviewed this manuscript with one of the reviewers who provided the listed reports.

Version 1:

Reviewer comments:

Reviewer #1

(Remarks to the Author)
No additional comments

Reviewer #2

(Remarks to the Author)
The majority of the comments were addressed to a satisfactory standard.

Reviewer #3

(Remarks to the Author)

Reviewer #5

(Remarks to the Author)
I co-reviewed this manuscript with one of the reviewers who provided the listed reports. This is part of the Nature Communications initiative to facilitate training in peer review and to provide appropriate recognition for Early Career Researchers who co-review manuscripts.

Dear editor and reviewers,

Thank you for your insightful commentary regarding this work; we believe that the critiques provided have prompted us to substantially strengthen our manuscript. We have performed several experiments in response to reviewers, and have incorporated several of these into the revised manuscript. We have used the Track Changes function in the revised manuscript to highlight where these changes were made. Here we provide a point-by-point response to all comments from each reviewer:

Reviewer #1 (Remarks to the Author):

The following points should be addressed:

1. The title is quite broad and doesn't reflect the new and exciting data. Authors might consider a revision.

We have re-assessed our title and agree with Reviewer #1; we have changed our title to reflect this.

2. At the end of the first paragraph, line 111, authors might consider a summary statement or two highlighting the results presented above and emphasizing what is new and what is confirmatory.

We thank Reviewer #1 for pointing out the lack of clarity here. The line referenced by the reviewer is at the end of the second paragraph in the Results section. We have added two sentences here, one in which we summarize these findings, and another in which we explain why sodium chlorate (NaClO₃) was used as our HS inhibition method of choice for subsequent experiments (revised submission, lines 114-121).

3. An enormous amount of data is presented and care needs to be taken to be sure adequate controls are well described.

We agree with the reviewer that detailed descriptions of controls used in each experiment are critical, and have revisited all statements regarding experimental controls in our initial submission. For western blotting data for the experiments focusing on Areg stimulation with/without HS inhibition in multiple cell types (**initial submission, Fig. 1 and Fig. S1**), we used EGF as a non-heparin binding control for experiments with Areg stimulation (initial submission, line 95-96), which has been previously used in the field of Areg research (Willmarth et al. 2009, *Cell Signal*; Fukuda et al. 2016, *Sci Rep*; von Joest et al. 2022, *Cell Rep*). For our RNA-seq experiment involving Areg stimulation of Col14-LMC with/without HS inhibition, we included a DEG comparison between vehicle and NaClO₃ treatment alone (i.e., no Areg treatment) to account for gene expression induced by this treatment (**initial submission, Fig. 2**), and have discussed this in the manuscript text (initial submission, line 177-178). For the KO cell line panel (**initial submission, Fig. 3**), we used WT LLC cells as controls for initial exploratory experiments into effects on Areg stimulation potential. In these experiments, to account for potential differences in EGFR levels between clonal KO cell lines, we calculated Areg signaling relative to EGF (which could be considered an additional “control” in this context). This was stated in the figure legend, and our intent was to further make this clear by the inclusion of sample EGF and Areg blots underneath graphs for each separate KO line; however, we have added a line to the revised manuscript upon the first instance of this experimental setup to further clarify for readers (revised submission, lines 239-240). In addition, in this section we found one instance where we described a set of experimental controls as “EGF controls” when it should say “WT controls” (initial submission, line 257) – we have corrected this error (revised submission, line 276). For follow-up experiments, we used an LLC cell line transfected with empty vector (i.e., the PX458 plasmid without gRNA inserts) as a control (**initial submission, Fig. 3G and Fig. S4G**), and reconfirmed results from the KO panel with this empty vector control cell line – this was described in the Supplementary Methods (initial submission, line 1219) and in the figure captions for **initial submission Fig. 3G and Fig. S4G**. For siRNA experiments (**initial submission, Fig. 4**), we believe our use of non-targeting siRNA was the most appropriate control, which we described in initial submission lines 278-279. For in vivo experiments and harvest of Col14-LMC for ex vivo experiments (**initial submission, Fig. 5/ Fig. S6/ Fig. S7/ Fig. 6**), we used Col1a2-CreER⁻ tamoxifen-treated littermate mice as controls. While corn oil-only controls for experiments with tamoxifen are sometimes utilized, in our view, the more appropriate control is to use tamoxifen with CreER-negative animals, to capture any side effects of tamoxifen treatment. For several of the experiments in this section, in addition to tamoxifen-treated, Col1a2-CreER⁻, IAV-

infected controls, we have also utilized tamoxifen-treated, mock-infected PBS controls to establish the baseline of our parameters in uninfected mice. To clarify that Col1a2-CreER⁻, tamoxifen-treated mice were used as the relevant controls for experiments here, we have clarified several instances of usage in the Results section describing these experiments (revised submission, lines 318, 324, 328-329, 346-348, 352).

Referee #2 (Remarks to the Author):

The work presented by Loffredo et al. demonstrates that heparan sulphate expressed on lung mesenchymal cell subpopulations plays important beneficial roles in regulating signalling responses to Treg cell secreted amphiregulin during influenza A infection. The manuscript is well written and follows a sound and logical progression of experiments that comprehensively demonstrate the importance of heparan sulphate. In my opinion, the experiments and data produced are sufficient to support the conclusions.

To further strengthen Figure 5/S5/S6 experimental evidence and draw conclusions, it would be beneficial to see the histopathology and immunochemical staining of the harvested lung tissues to get an idea of the extend of expression and localization of the evaluated cell types/HS, and the consequences of lung tissue inflammation/structural deterioration and fibrosis after viral load and bleomycin treatments.

We agree that histological analyses are highly necessary in this context; specific lobes of the lungs from the IAV experiments shown in our original manuscript were processed for histology. We have worked with a board-certified pathologist with expertise in lung pathology at our institution (Dr. Anjali Saqi, now included as a co-author in our revised manuscript) to help perform these analyses.

Initially, we sought to investigate the percentage of inflamed area in the lungs of control or HS^{ckO} IAV-infected lungs, using hematoxylin and eosin (H&E) staining and quantification of inflamed areas; we saw no changes in this parameter (**Reviewer Fig. 1A**). This is consistent with the largely unchanged immune infiltrate profile for these lungs, as reported in our first submission (**initial submission, Fig. S7**). We additionally attempted to analyze airway/alveolar damage using these stains. The small airways/bronchioles in both control and HS^{ckO} IAV-infected lungs appear to contain essentially intact epithelial barriers, with minimal chronic inflammation of airway epithelium and little apparent sloughing of epithelial cells at this timepoint (**Reviewer Fig. 1B**). Assessment of alveolar epithelial cells in affected alveolar areas of both control and HS^{ckO} IAV-infected lungs is almost completely obscured by the immune infiltrate (**Reviewer Fig. 1B**); thus, histological analysis of damage in alveolar cell types was not able to be performed in this fashion. However, we cannot rule out that histological changes in epithelial cells may be more pronounced at later timepoints, potentially related to increased dysplastic repair/Krt5⁺ pod formation, which was a consequence of Treg cell Areg deficiency during this repair context in our previous study (Kaiser et al. 2023, *J Exp Med*). We also performed Masson's Trichrome staining on lung sections to ascertain if there were differences in fibrosis during the IAV model (i.e., blue staining in alveolar areas, separate from the expected normal peri-bronchiolar and perivascular blue staining also present in uninfected mice), and saw no apparent visual changes (**Reviewer Fig. 1C**); fibrosis resulting from the IAV model has been shown to occur at later timepoints following recovery from initial infection (Keeler et al. 2018, *J Immunol*). The primary focus of our study was to investigate the role of Treg cell-derived Areg and HSPG on mesenchymal cells at early timepoints following IAV infection.

Reviewer Figure 1

A. Quantification of inflamed areas from H&E-stained lung sections:

B.

PBS (no IAV)

H&E stain
Control (IAV-infected)

HS^{CKO} (IAV-infected)

40x insets:
PBS (No IAV)

Control (IAV-infected)

HS^{CKO} (IAV-infected)

C.

Masson's trichrome stain, 40x zoom
PBS (No IAV)

Control (IAV-infected)

HS^{KO} (IAV-infected)

Reviewer Figure 1

A) Left: example of quantification of inflamed full mouse lung lobes, with yellow outlines highlighting inflamed regions following IAV infection (8 d.p.i.). Right: Percentage of inflamed area in mock-infected PBS-treated, control (Col1a2-CreER⁻) IAV-infected mice, and HS^{ckO} IAV-infected mice. Experiment was performed 3 times; all values are included in graph. Standard error displayed on graph; n.s.: not significant. **B)** Top: Full lobes of hematoxylin and eosin (H&E)-stained lungs from PBS-treated (no IAV), control (Col1a2-CreER⁻) IAV-infected, and HS^{ckO} IAV-infected mice, with zoomed insets highlighted with black boxes. Bottom: 40x zoomed-in insets on representative inflamed area from lungs of each treatment/genotype stained with H&E, to highlight airway and alveolar areas. **C)** 40x zoomed-in sections on representative inflamed area from lungs of each treatment/genotype stained with Masson's Trichrome, to highlight lack of fibrosis in alveolar parenchyma (blue).

To address potential changes to the lung epithelium that would reflect deficient tissue repair using an alternative method (also related to a question from Reviewer 3), we also performed flow cytometry on the epithelium in these experiments, identifying upper airway cells (i.e., ciliated, club, and basal cells) vs. alveolar cells (i.e., AT1 and AT2 cells) based on the expression of CD24 and CD104 (integrin $\beta 4$) as others have previously reported (Chen et al. 2012, *Stem Cells*; Cassandras et al. 2020, *Nat Cell Biol*) (**Reviewer Fig. 2A**). We found that there were significant differences in the amount of live upper airway cells vs. alveolar cells recovered from control vs. HS^{ckO} IAV-infected lungs, reflected by a decrease in the proportion of recovered alveolar epithelial cells (**Reviewer Fig. 2B**), an increase in the proportion of recovered upper airway cells (**Reviewer Fig. 2C**), and a decrease in the ratio of alveolar cell to upper airway cell counts (**Reviewer Fig. 2D**). Given that there appeared to be few alterations in upper airways based on histology (**Reviewer Fig. 1C**), these metrics are likely an indicator of fewer live alveolar cells recovered from HS^{ckO} IAV-infected lungs compared to control IAV-infected lungs, which would reflect increased alveolar epithelial damage in this context. When coupled with our blood oxygen saturation readout of lung function, as well as the decrease in the proliferation of reparative mesenchymal cells, we believe this provides further evidence for deficient epithelial repair in the HS^{ckO} context.

Reviewer Figure 2

A.

B.

C.

D.

Reviewer Figure 2

A) Gating strategy for analysis of upper airway vs. alveolar epithelial cells from lungs. **B-D)** Proportions of live alveolar epithelial cells (**B**), or upper airway cells (**C**) of total epithelial cells (CD45⁻CD31⁻Epcam⁺) recovered from lung preparations for control (Col1a2-CreER⁻) IAV-infected mice and HS^{ckO} IAV-infected mice (8 d.p.i.). Graph in (**D**) reflects ratio of alveolar epithelial cell counts to upper airway cell counts. Experiments were performed 3 times; all values are included in graphs. Standard error displayed on graphs; n.s: not significant, *: 0.01<p<0.05, **: 0.001<p<0.01, ***: 0.0001<p<0.001, ****: p<0.0001.

The data presented above as **Reviewer Figures 1 and 2** have now been added to the revised manuscript (now appearing as **revised manuscript Fig. S9**), with associated descriptions in Results (revised submission lines 361-385) and updated Methods (revised submission lines 608-615, 1291-1295), as we agree that these are important analyses for interpretation of our presented results.

In addition, to address this and Reviewer 3's points about expression and localization of the evaluated cell types, we have performed additional experiments using immunofluorescence staining of lung tissue during the IAV model, in control and HS^{ckO} mice. We stained for Col14-LMC (using previously established marker Entpd2 from Kaiser et al. 2023, *J Exp Med*), HS (using the 10E4 antibody), and Ki-67 in this context in an attempt to answer these questions. Firstly, as expected, we found that Ki-67 is widely distributed throughout lung tissue following IAV infection (**Reviewer Fig. 3A**). However, it is difficult to discern from this staining whether the positive Ki-67 staining is present on infiltrating immune cells or on structural cells such as Col14-LMC. Using various analysis methods, we attempted to quantify the total number of Ki-67⁺ cells within Entpd2⁺ areas (i.e., areas with Col14-LMC): 1) determination of % Ki-67⁺ pixels in manually outlined Entpd2⁺ areas (since Entpd2⁺ staining is discontinuous/patchy) (**Reviewer Fig. 3B**); 2) determination of Ki-67⁺ cells by particle analysis in manually outlined Entpd2⁺ areas (**Reviewer Fig. 3C**); or 3) colocalization analysis of % Ki-67⁺ pixels in total Entpd2⁺ areas (**Reviewer Fig. 3D**). None of these analysis methods showed changes between control IAV-infected and HS^{ckO} IAV-infected mice. We believe that due to the inherent complication of quantifying Ki-67 from immunofluorescence when there are many proliferating immune cells present, and since FACS analysis allows for isolation of all Col14-LMC in the digested lung in this context, our original approach to quantify Ki-67 on Col14-LMC via FACS (**initial submission, Fig. 5E**) is the more appropriate method for this evaluation. With regards to HS staining in lung tissue, HS was found to be widely distributed throughout the lung on multiple cell types using the 10E4 antibody for immunofluorescence staining of lung sections – which is reflective of HS being expressed on multiple lung cell types (as shown in **initial submission, Fig. 5A–B**)

Reviewer Figure 3

A.

Reviewer Figure 3

A) Representative immunofluorescence staining of DAPI (blue, not included in composite image), HS (10E4 antibody) (green), Ki67 (red), and Entpd2 (white), in control IAV-infected mouse lung (7 d.p.i.) **B)** Percentage of Ki67⁺ pixels in manually outlined Entpd2⁺ areas of control (Col1a2-CreER⁻) and HS^{ckO} IAV-infected mice (7 d.p.i.). **C)** Quantification of Ki67⁺ cells by particle analysis in manually outlined Entpd2⁺ areas of control (Col1a2-CreER⁻) and HS^{ckO} IAV-infected mice (7 d.p.i.). **D)** Percentage of colocalized Ki67⁺ pixels in total Entpd2⁺ areas of control (Col1a2-CreER⁻) and HS^{ckO} IAV-infected mice (7 d.p.i.). Experiments were performed 2 times; all values are included in graphs. Standard error displayed on graphs; n.s: not significant.

I note that the description given in the methods seems to suggest that the entire lungs were digested for cell analysis.

For the IAV experiments, different parts of the lungs – matched between animals – were taken for FACS analysis, RNA analysis of viral load, and histological analysis (revised submission lines 608-610). Entire lungs were used for processing for the bleomycin experiments (initial submission line 557), which can be more susceptible to stochastic distribution of the damaging agent (bleomycin) upon administration than an agent such as viruses that can spread within the lung.

Would it be possible to reference the studies of the Farber lab regarding IAV infection and KEC infection assays?

We have consulted with the Farber lab and referenced the study where this work is described (Teijaro et al. 2010, *J Virology*) (revised submission line 583).

Some minor formatting issues that were detected during the review, although that is not to say all typos/errors were identified. Please thoroughly check the manuscript as necessary.

1. In abstract line 20, heparan sulfate is written in full despite that previously the abbreviation was given. The full name of IAV, which includes “virus” is not given in abstract line 22.

We thank the referee for bringing these errors to our attention and have revised the abstract to correct these concerns.

2. Fig. 1B is called out twice, in two separate paragraphs. It is suggested that these two instances are combined into one paragraph, for ease of reading.

Initial submission Fig. 1B is used to describe first the lack of EGFR phosphorylation in the context of HS inhibition (initial submission line 98), then separately to describe the overall maintained nature in ERK phosphorylation and only partial reduction in AKT phosphorylation (initial submission line 117). Since these are

separate conceptual discussions, we felt it important to call out **initial submission Fig. 1B** in each of these locations; however, we would be happy to incorporate this change for clarity if deemed necessary.

3. Line 299, switch order of “maximum” and “the”.

This correction has also been incorporated. Thank you.

4. Line 353, additional “in” can be deleted.

This correction has also been incorporated. Thank you.

Referee #3 (Remarks to the Author):

In the paper “Heparan sulfate regulates amphiregulin signaling towards reparative lung mesenchymal cells during influenza A infection” from Loffredo et al., the authors described heparan sulfate (HS) as new regulator of AREG signaling in Col14-LMC in vitro and during IAV infection in mice. They concluded that Treg-derived AREG induced reparative signaling in Col14-LMC during IAV infection in a HS-dependent manner. The study is well done with appropriate methods and controls. The paper is well written and the data are presented in an understandable way. It is a follow-up study based on previous publications of the group. Here, the authors described a novel mechanism by which the effect of AREG on lung tissue cells can be regulated during lung infection.

However, I have some major concerns regarding the conclusion, which I think has not been properly shown. The only repair parameter the authors analyzed after IAV infection was Ki-67 in isolated Col14-LMC. No data from lung tissue were shown, neither data that would allow a correlation of Ki-67 expression with AREG or Treg-derived AREG in the mouse model.

Major points:

Figure 5: The authors showed a reduced proliferation of Col14-LMC after viral infection in HS^{CKO} mice. Did they also observe altered AREG/HS downstream signaling in Col14-LMC from these mice?

Downstream signaling is difficult to experimentally evaluate from cells taken directly from lungs after the processing inherent in FACS experiments. Thus, the experiments done in the earlier parts of the manuscript, with direct AREG treatment of primary Col14-LMC cells in which HS has been inhibited (mimicking HS^{CKO} mice), provides a better look at these downstream signaling modalities. We established in these experiments that in Col14-LMC with HS inhibition, AREG can still stimulate downstream signaling, but of a different nature than with HS intact (**initial submission, Fig. 1F**), further reflected by the altered transcriptional programs (Hippo signaling, actin mobilization, intermediate filament induction, and gap junction formation) that are induced by Areg in HS-intact, but not HS-inhibited Col14-LMC (**initial submission, Fig. 2**).

However, we understand the concerns of the reviewer, in that we still do not know the downstream consequences of Areg signaling on Col14-LMC that are allowing them to affect tissue repair modalities. The specific nature by which certain mesenchymal cells signal towards the epithelium for repair is still an area of active investigation, with fibroblast growth factors (FGFs) and Wnt mediators having been previously identified in this context (El Agha & Thannickal 2023, *J Clin Invest*). We want to profoundly thank Reviewer 3 for highlighting this deficiency in our manuscript and prompting us to revisit this, as we feel that not showing an exploration of FGF and Wnt expression within our RNA-seq data would have left a notable gap in our story, given the importance of these mediators in the field. In **Reviewer Fig. 4**, we highlight all FGF and Wnt genes from our RNA-seq dataset that are expressed on average >2 TPM, in the same manner as **initial submission Fig. 2**. Notably, *Fgf1*, *Fgf18*, *Wnt2*, and *Wnt2b* show similar patterns to the HS-dependent genes from **initial submission Fig. 2** – substantial upregulation by Areg treatment in the HS-intact scenario, and lessened or absent upregulation upon HS inhibition. While this same pattern is not apparent for all FGF and Wnt genes, the HS-dependent nature of certain of these genes may point to a specific set of reparative mediators induced by Areg that require HS on target cells for proper induction. As we feel this is a critical point for readers, we have included **Reviewer Fig. 4** as a new supplementary figure in the main paper, **revised submission Fig. S3**, with accompanying descriptions in

Results (revised submission lines 212-220) and Discussion (revised submission lines 443-444). We have changed the numbering of subsequent supplementary figures accordingly throughout the manuscript.

Reviewer Figure 4

Reviewer Figure 4

Transcript per million (TPM) values from RNA-seq for all FGF genes and Wnt genes with average TPM>2; significance across samples in each group (n=3) calculated by Kruskal-Wallis test. Standard error displayed on graphs; n.s.: not significant, *: 0.01<p<0.05, **: 0.001<p<0.01, ***: 0.0001<p<0.001, ****: p<0.0001.

We also explored in this context the IL-6 family cytokine LIF, which shows substantial upregulation in Col14-LMC in our RNA-seq dataset upon AREG exposure (**initial submission, Fig. 2**) and in our co-culture experiments upon IAV-induced Treg cell exposure (**initial submission, Fig. 6**). LIF has previously been shown to be critical for lung protection and/or repair in the context of bacterial pneumonia infection (Quinton et al. 2012, *J Immunol*), and plays recently defined roles in the context of immune cell trafficking during viral infection (Gogoi et al. 2024, *Nature*). We assessed LIF production at the protein level by FACS in the broad cell types in the lung, as well as specific LMC subsets (**Reviewer Fig. 5A**). Of the broad cell types in the lung, there was apparent production by all groups, but only mesenchymal cells and endothelial cells showed significant upregulation of LIF upon IAV infection, compared to PBS-treated controls. Strikingly, Col14-LMC are the only LMC subset that significantly upregulates LIF upon IAV infection, with this increase being of higher magnitude than the increase in endothelial cells. Thus, Col14-LMC are an unheralded source of increased LIF production in the context of IAV infection. We further attempted to assess LIF induction in Col14-LMC in HS^{CKO} mice in the context of IAV infection (**Reviewer Fig. 5B**). We did not see a change in LIF induction upon IAV infection between control IAV-infected and HS^{CKO} IAV-infected mice. However, it must be noted that the antibody staining strategy used here involved intracellular staining coupled with the use of brefeldin A (to block ER to Golgi transport) and monensin (to block Golgi trafficking); while we felt this was the best strategy to utilize in order to assess a secreted cytokine such as LIF

at the protein level by FACS, it may be overestimating the amount of LIF actually produced in this context. A genetic fluorescent reporter strategy could potentially be more informative for this assessment. As this is an ongoing area of research for our laboratory, we have not added these results to the manuscript, but would be happy to do so if deemed necessary.

Reviewer Figure 5

Reviewer Figure 5

A) LIF production, assessed by FACS, on indicated cell populations from mock-infection control (PBS) mice (n=6) and IAV-infected (100 TCID₅₀) mice at 7 d.p.i. (n=6). Graphs contain all values from 2 separate experiments. **B)** LIF production, assessed by FACS, on indicated cell populations from mock-infection control (PBS) mice (n=3), IAV-infected (100 TCID₅₀) control mice (n=10), and IAV-infected HS^{ckO} mice (n=12) at 7 d.p.i. (all tamoxifen-treated). Graphs contain all values from 2 separate experiments. Gating strategies shown in initial submission Fig. S2 (total lung). Gating based on polyclonal rabbit IgG control. Lung digests were treated with brefeldin A and monensin during digestion, and staining was done intracellularly. Standard error displayed on graphs; n.s: not significant, *: 0.01<p<0.05, **: 0.001<p<0.01, ***: 0.0001<p<0.001, ****: p<0.0001.

The authors showed SpO₂ as only disease parameter. If the lung tissue is damaged, they should show this in histological stainings. Was the lung tissue damage altered in HS^{ckO} compared to control mice?

We thank the reviewer for their comment and have addressed histological differences and additional disease parameters (i.e., a decrease in live alveolar epithelial cells recovered from harvested lungs) in response to a similar query from Reviewer 2 (please see above, Reviewer Fig. 1 and Reviewer Fig. 2).

Did Col14-LMC express Ki67 in lung tissue of HS^{ckO} mice and was Ki-67 expression localized to areas of tissue damage/repair?

As indicated in initial submission Fig. 5E, we do see an appreciable upregulation of expression of Ki-67 even in HS^{ckO} lung tissue undergoing IAV infection (compare to PBS-treated controls). As to its localization to areas of tissue damage/repair, we attempted to address this using the immunofluorescence approach described in the response to a similar query by Reviewer 2. By immunofluorescence assessment of IAV-infected lungs (Reviewer Fig. 3), we found that Ki-67 expression is widespread throughout the tissue, with a large presence on infiltrating immune cells in areas of tissue damage.

Were AREG levels altered in HS^{ckO} mice compared to control mice? Did Tregs express AREG in HS^{ckO} mice and localize near Col14-LMC?

We included staining for AREG in lung immune cells by FACS in our IAV experiments – there are no changes in AREG production levels in lymphoid or myeloid cells overall (Reviewer Fig. 6A-B), or specifically by Treg cells (Reviewer Fig. 6C) in HS^{ckO} mice compared to control mice. Consistent with our model, wherein AREG production is upstream of HS-related signaling mediation differences, we did not expect to see difference in AREG production in the HS^{ckO} mice (in other words, we attribute our effects here not to different amounts of AREG in the local environment, but to the ability of receiver Col14-LMC cells to integrate these signals). As we

agree this is an important point to consider, the data presented in **Reviewer Fig. 5C** have now been included within **revised submission Fig. S7**, with associated descriptions in Results (revised submission line 356), Discussion (revised submission lines 500-501), updated figure legend (revised submission lines 1075-1077), and updated Methods (revised submission lines 1284-1285).

Reviewer Figure 6

Reviewer Figure 6

A-C Percent AREG⁺ staining of total myeloid cells (**A**), total lymphoid cells (**B**), or Treg cells (**C**) in PBS-treated (no IAV), control (Col1a2-CreER⁻) IAV-infected, and HS^{ckO} IAV-infected mice (8 d.p.i.). Experiments were performed 3 times; all values are included in graphs. Gating strategies shown in **initial submission, Fig. S7**. Standard error displayed on graphs; n.s.: not significant.

The reviewer also asks if Treg cells localize near Col14-LMC – imaging experiments showing preferential localization of Treg cells to areas near Col14-LMC and to areas of AT2 cell denudation (i.e., damaged lung regions) during IAV infection were demonstrated in our previous publication (Kaiser et al. 2023, *J Exp Med*).

Was Ki-67 expression of Col14-LMC altered after blockage of AREG? Did the mice develop more severe lung tissue damage?

We assessed the effect of AREG blocking on Col14-LMC Ki-67 levels – we did not see an expected reduction (**Reviewer Fig. 7**). We note that this is a different experimental setup than the HS^{ckO} experiments where we previously saw this phenotype, involving the daily treatment of a control or anti-AREG antibody at high doses – perhaps these treatments are affecting the tissue in some other way that is altering these results. More work is needed on experimental antibody-mediated blocking of Areg activity to ascertain why we see a significant difference in mouse blood oxygen saturation upon treatment without corresponding loss of proliferative activity of Col14-LMC.

Reviewer Figure 7

Reviewer Figure 7

Percent Ki-67⁺ staining of Col14-LMC from control (IgG-treated) IAV-infected, and αAreg-treated IAV-infected mice (8 d.p.i.). Experiments were performed 3 times; all values are included in graphs. Standard error displayed on graphs; n.s.: not significant.

Why did bleomycin-treated HS^{CKO} mice show no differences in SpO₂ compared to control mice? The authors should at least discuss this finding.

In this work, we did not explore SpO₂ differences in mice during the bleomycin model. We believe the reviewer may be referring to data presented in **initial submission Fig. S6E**, showing weight loss data during the bleomycin model in control and HS^{CKO} mice, or **initial submission Fig. S6F**, showing the expression of alpha-smooth muscle actin (α SMA) on LMC during the bleomycin model in control and HS^{CKO} mice – a different readout for pathology in this context, focusing on fibrosis. SpO₂ as an indicator of bleomycin-induced pathology is not routinely used in this model, and it is not known whether Treg cell-derived Areg has any effect on SpO₂ in this context. However, readouts of fibrosis are at this time a more established indicator of pathology during the bleomycin model. This is the basis for our claim that while HS-mediated Areg signaling to Col14-LMC functionally influences repair in the setting of viral lung damage, this signaling relay may be specific to the type of lung pathology – since we do not observe an effect on the induction of fibrosis (in the bleomycin model).

Figure 6: Protein expression analysis of Il6 and Vegfa would strengthen the gene expression data. It is not clear why these three genes were analyzed. Are they involved in regenerative/repair processes of Col14-LMC in lung disease? Is expression of Ki-67 induced/alterd in this experimental setup?

These genes have previously been shown by our group to be highly upregulated in Col14-LMC upon exposure to AREG (Kaiser et al. 2023, *J Exp Med*), and upon exposure to activated lung Treg cells themselves (*Lif/Il6* only) (Kaiser et al. 2023, *J Exp Med*; Loffredo et al. 2024, *bioRxiv*). Previous work has shown the importance of these factors during recovery from lung disease, although the specific cellular sources of these factors are either unclear or alternative to Col14-LMC in this previous work (Yang et al. 2017, *Sci Rep*; Quinton et al. 2012, *J Immunol*; Vila Ellis et al. 2020, *Dev Cell*). As demonstrated for LIF in **Reviewer Fig. 5** above, Col14-LMC may represent an as-yet unappreciated source for these factors in lung repair. As for analyzing these genes at the protein level in this experimental setup, we appreciate that these data would be helpful; however, this assay necessitates amounts of Col14-LMC (40,000–50,000 per well) and Treg cells (20,000–25,000 per well) that are difficult to harvest due to their paucity. To be able to accurately harvest protein for western blotting, we would need to drastically increase the number of mice harvested for these experiments. For this reason, we also feel it would be difficult to analyze Ki-67 expression at the protein level. However, we have looked at *Mki67* expression at the RNA level in this experimental setup, and found that – similar to *Lif*, *Il6*, and *Vegfa* – *Mki67* is significantly increased in Col14-LMC upon Treg cell co-culture, but not increased (and in fact significantly lower at baseline) in HS^{CKO} Col14-LMC upon Treg cell co-culture (**Reviewer Fig. 8**).

Reviewer Figure 8

Reviewer Figure 8

qPCR for *Mki67* from control (n=6) and HS^{CKO} (n=6) Col14-LMC without inclusion of IAV-induced lung Treg cells, or control (n=6) and HS^{CKO} (n=6) Col14-LMC with inclusion of IAV-induced (300 TCID₅₀) lung Treg cells. Expression values computed as copies of target gene per 10,000 copies of housekeeping gene (Hprt). Expression values were normalized across experiments. Graphs contain all values from 2 experiments. Standard error displayed on graphs; n.s.: not significant, *: 0.01 < p < 0.05, **: 0.001 < p < 0.01, ***: 0.0001 < p < 0.001, ****: p < 0.0001.

Minor points:

The authors showed three values per group and in the figure legend they stated “n=3 per condition, graphs contain all values from 3 separate experiments”. I think that the mean values of three experiments are depicted. However, in Fig. 1C, they showed 3-4 values derived from two experiments. This needs to be clarified.

We sought to include the complete sample size (n=) and the number of experiments run for each separate graph/set of graphs shown throughout the manuscript. In **initial submission Fig. 1C**, we performed the FACS experiment two times, with n=3-4 total per group – this is reflected in the number of points on the graphs for this figure. We believe the “n=3 per condition, graphs contain all values from 3 separate experiments” statement that Reviewer 3 references refers to data in **initial submission Fig. 1B**, which are analyzed based on quantifications of three separate western blot experiments – not the FACS experiments in Fig. 1C, wherein individual biological replicates are plotted.

The authors used a high concentration of murine AREG (500 ng/ml) in vitro to stimulate the cells. I wonder if this high concentration is necessary to see an effect of AREG; how can these data be transferred to the in vivo situation where Tregs (always low in number) will never produce such high AREG levels?

While we agree with this reviewer that the levels of rmAREG used in this study are high, we have considered several factors in the rationale of using this concentration. First, we performed initial titration experiments to determine the concentration of rmAREG that promotes similar EGFR phosphorylation to the standard rmEGF concentration used in the literature (100 ng/ml). Secondly, for proteins such as AREG that may be operating via close-range paracrine mechanisms between provider cells and target cells, the local concentration in the interface between these cells is difficult to measure, and is likely not accurately captured by methods reliant on full tissue digestion. Indeed, the effect of HS on AREG that we study here could be related to its ability to sequester AREG near its receptor, which would increase the effective local concentration substantially. Third, in one of our most critical experiments with recombinant AREG herein (i.e., the RNA-seq on Col14-LMC) (**initial submission, Fig. 2**), we have tested a range of AREG concentrations to account for this.

Figure 2: Why is it necessary to include nearly the same number of DEGs in vehicle treated Col14-LMC and NaClO₃-treated Col14-LMC for KEGG analysis? To achieve this, the authors compared Col14-LMC that were cultured with different AREG concentrations (200 ng/ml vs. 1000 ng/ml). It is known that the effect of AREG on other cells depends on the concentration used for the in vitro experiments. Are the results the same, if Col14-LMC treated with the same AREG concentration are compared?

We understand that our approach for this analysis (**initial submission, Fig. 2**) may merit further explanation. While we show in **initial submission Fig. 1 and Fig. S1** that Areg signaling can still occur upon HS inhibition or in cells completely lacking HS, from these experiments it also appears that signaling is occurring at a lower level for EGFR phosphorylation and AKT phosphorylation in HS-inhibited cells than in HS-intact cells. It could be argued that this is further manifested by lesser gene expression differences upon treatment of HS-inhibited Col14-LMC with rmAREG treatment compared to HS-intact Col14-LMC in **initial submission Fig. 2**. However, since ERK downstream signaling appears to be largely intact from these experiments, we propose in our original manuscript that the qualitative nature of the downstream signaling here is different in ways that cannot simply be explained by “more” vs. “less” signaling. The potential ability of HS to sequester Areg in proximity to EGFR on the surface of target cells may alter the effective concentration of Areg, as we argued in an earlier point. This was the basis for including a range of concentrations in this experiment, and for showing the full range of concentrations for a subset of genes. For the pathway analysis that Reviewer 3 has brought up here, the goal was to probe pathways activated at roughly equivalent signaling levels (with significantly upregulated genes as a proxy), which is why we performed this analysis with different levels of AREG stimulation for HS-intact vs. HS-inhibited scenarios.

Regardless, we have also run the analysis in the way suggested by the reviewer (**Reviewer Fig. 9, left**); we also re-ran the analysis from **initial submission Fig. 2** and included it side-by-side for comparison (**Reviewer Fig. 9, right**). Note that, likely due to updates to the gProfiler databases since we ran our original analysis, “Efferocytosis” was now included as a significantly altered pathway in both HS-intact and HS-inhibited scenarios in our original analysis style (right), with this being the only difference resultant from these updates from our original implementation of gProfiler in January 2023 (**initial submission, Fig. 2C**); the significance of this is unclear, since mesenchymal cells in the lung are not thought to participate in appreciable efferocytosis. Comparing the newly run analysis (both HS-intact and HS-inhibited treated with 200 ng/ml rmAREG) to our original analysis (HS-intact treated with 200 ng/ml, HS-inhibited treated with 1000 ng/ml), as may be expected,

we find that fewer processes are represented in the HS-inhibition scenario, likely as a consequence of less upregulated genes in this context. However, the “MAPK signaling pathway”, which we showed to be largely maintained in HS-inhibited cells by western blotting (**initial submission, Fig. 1 and Fig. S1**), is still significantly altered in both scenarios, recapitulating these results. The “ErbB signaling pathway” appears as significant in both HS-intact and HS-inhibited in the new analysis (left), which is curious as Areg treatment at a higher concentration (500 ng/ml) was clearly deficient for stimulating EGFR phosphorylation in HS-inhibited cells in western blotting experiments (**initial submission, Fig. 1 and Fig. S1**). This may be related to alternative EGFR phosphorylation sites than the one tested here (Y1068) being activated in the context of Areg signaling to HS-inhibited cells, which is an interesting avenue of research but not one that we pursued in this work. Overall, these analyses are largely similar to those performed in **initial submission Fig. 2**.

Reviewer Figure 9

Reviewer Figure 9

Left: Pathway analysis (gProfiler, KEGG pathways) on upregulated genes from RNA-seq, from the “No HS inhibition, vehicle vs. AREG (200 ng/ml)” comparison (1459 DEGs) (black circle), and the “HS inhibition, vehicle vs. AREG (200 ng/ml)” comparison (336 DEGs) (red circle). **Right:** Pathway analysis on upregulated genes from RNA-seq from the “No HS inhibition, vehicle vs. AREG (200 ng/ml)” comparison (1459 DEGs) (black circle), and the “HS inhibition, vehicle vs. AREG (1000 ng/ml)” comparison (1285 DEGs) (red circle). KEGG pathways shown are significant by Benjamini-Hochberg FDR adjusted p-value (<0.05). Analysis was ran in September 2024.

In Fig. 4, the authors used the term bleomycin infection. I think this is not correct because bleomycin is a substance and not a virus. In the Figure legend of S5C, the authors also wrote bleomycin-treated (14 d.p.i.). If p.i. stands for post-infection, this is also not correct.

We thank the reviewer for bringing this error to our attention. The instance where we wrote “bleomycin infection” instead of “bleomycin treatment” (initial submission line 287) has been corrected (revised submission line 308). For the “d.p.i.” concern, we found that we had not properly introduced the acronym, and thus inserted this definition at revised submission line 351-352 (the first instance of use in the manuscript). We generally define “d.p.i.” as “days post-inoculation” (and have done so here), especially when working with both the IAV and bleomycin models so that the nomenclature is similar between these models.

References for point-by-point response to reviewers:

Cassandras M, Wang C, Kathiriya J, Tsukui T, Matatia P, Matthay M, Wolters P, Molofsky A, Sheppard D, Chapman H, Peng T. Gli1⁺ mesenchymal stromal cells form a pathological niche to promote airway progenitor metaplasia in the fibrotic lung. *Nat Cell Biol.* 2020 Nov;22(11):1295-1306. doi: 10.1038/s41556-020-00591-9. Epub 2020 Oct 12. PMID: 33046884; PMCID: PMC7642162.

Chen H, Matsumoto K, Brockway BL, Rackley CR, Liang J, Lee JH, Jiang D, Noble PW, Randell SH, Kim CF, Stripp BR. Airway epithelial progenitors are region specific and show differential responses to bleomycin-

- induced lung injury. *Stem Cells*. 2012 Sep;30(9):1948-60. doi: 10.1002/stem.1150. PMID: 22696116; PMCID: PMC4083019.
- El Agha E, Thannickal VJ. The lung mesenchyme in development, regeneration, and fibrosis. *J Clin Invest*. 2023 Jul 17;133(14):e170498. doi: 10.1172/JCI170498. PMID: 37463440; PMCID: PMC10348757.
- Fukuda S, Nishida-Fukuda H, Nanba D, Nakashiro K, Nakayama H, Kubota H, Higashiyama S. Reversible interconversion and maintenance of mammary epithelial cell characteristics by the ligand-regulated EGFR system. *Sci Rep*. 2016 Feb 2;6:20209. doi: 10.1038/srep20209. PMID: 26831618; PMCID: PMC4735799.
- Gogoi M, Clark PA, Ferreira ACF, Rodriguez Rodriguez N, Heycock M, Ko M, Murphy JE, Chen V, Luan SL, Jolin HE, McKenzie ANJ. ILC2-derived LIF licences progress from tissue to systemic immunity. *Nature*. 2024 Aug;632(8026):885-892. doi: 10.1038/s41586-024-07746-w. Epub 2024 Aug 7. PMID: 39112698; PMCID: PMC11338826.
- Kaiser KA, Loffredo LF, Santos-Alexis KL, Ringham OR, Arpaia N. Regulation of the alveolar regenerative niche by amphiregulin-producing regulatory T cells. *J Exp Med*. 2023 Mar 6;220(3):e20221462. doi: 10.1084/jem.20221462. Epub 2022 Dec 19. PMID: 36534084; PMCID: PMC9767680.
- Keeler SP, Agapov EV, Hinojosa ME, Letvin AN, Wu K, Holtzman MJ. Influenza A Virus Infection Causes Chronic Lung Disease Linked to Sites of Active Viral RNA Remnants. *J Immunol*. 2018 Oct 15;201(8):2354-2368. doi: 10.4049/jimmunol.1800671. Epub 2018 Sep 12. PMID: 30209189; PMCID: PMC6179922.
- Loffredo LF, Kaiser KA, Kornberg A, Rao S, de Los Santos-Alexis K, Han A, Arpaia N. An amphiregulin reporter mouse enables transcriptional and clonal expansion analysis of reparative lung Treg cells. *bioRxiv [Preprint]*. 2024 Sep 28:2024.09.26.615245. doi: 10.1101/2024.09.26.615245. PMID: 39386607; PMCID: PMC11463663.
- Quinton LJ, Mizgerd JP, Hilliard KL, Jones MR, Kwon CY, Allen E. Leukemia inhibitory factor signaling is required for lung protection during pneumonia. *J Immunol*. 2012 Jun 15;188(12):6300-8. doi: 10.4049/jimmunol.1200256. Epub 2012 May 11. PMID: 22581855; PMCID: PMC3370070.
- Teijaro JR, Verhoeven D, Page CA, Turner D, Farber DL. Memory CD4 T cells direct protective responses to influenza virus in the lungs through helper-independent mechanisms. *J Virol*. 2010 Sep;84(18):9217-26. doi: 10.1128/JVI.01069-10. Epub 2010 Jun 30. PMID: 20592069; PMCID: PMC2937635.
- Vila Ellis L, Cain MP, Hutchison V, Flodby P, Crandall ED, Borok Z, Zhou B, Ostrin EJ, Wythe JD, Chen J. Epithelial Vegfa Specifies a Distinct Endothelial Population in the Mouse Lung. *Dev Cell*. 2020 Mar 9;52(5):617-630.e6. doi: 10.1016/j.devcel.2020.01.009. Epub 2020 Feb 13. PMID: 32059772; PMCID: PMC7170573.
- von Joest M, Chen C, Douché T, Chantrel J, Chiche A, Gianetto QG, Matondo M, Li H. Amphiregulin mediates non-cell-autonomous effect of senescence on reprogramming. *Cell Rep*. 2022 Jul 12;40(2):111074. doi: 10.1016/j.celrep.2022.111074. PMID: 35830812.
- Willmarth NE, Baillo A, Dziubinski ML, Wilson K, Riese DJ 2nd, Ethier SP. Altered EGFR localization and degradation in human breast cancer cells with an amphiregulin/EGFR autocrine loop. *Cell Signal*. 2009 Feb;21(2):212-9. doi: 10.1016/j.cellsig.2008.10.003. Epub 2008 Oct 14. PMID: 18951974; PMCID: PMC2632975.
- Yang ML, Wang CT, Yang SJ, Leu CH, Chen SH, Wu CL, Shiau AL. IL-6 ameliorates acute lung injury in influenza virus infection. *Sci Rep*. 2017 Mar 6;7:43829. doi: 10.1038/srep43829. PMID: 28262742; PMCID: PMC5338329.